# True Multimodal In-Context Learning Needs Attention to the Visual Context

**Shuo Chen[1,3,5,6]\*, Jianzhe Liu[2]\*, Zhen Han[1], Yan Xia[4], Daniel Cremers[2,5], Philip Torr[7], Volker Tresp[1,5], Jindong Gu[7]†**

[1]LMU Munich, [2]Technical University of Munich, [3]Siemens AG
[4]University of Science and Technology of China,
[5]Munich Center for Machine Learning (MCML),
[6]Konrad Zuse School of Excellence in Reliable AI (relAI),
[7]University of Oxford

## Abstract

Multimodal Large Language Models (MLLMs), built on powerful language backbones, have enabled Multimodal In-Context Learning (MICL)—adapting to new tasks from a few multimodal demonstrations consisting of images, questions, and answers. Despite showing noticeable improvement on standard vision-language datasets, current MLLMs struggle to leverage visual information in the demonstrations. Specifically, they tend to neglect visual cues and over-rely on textual patterns, leading to mere text imitation rather than genuine multimodal adaptation. This behavior makes MICL still unimodal and largely restricts its practical utility. More importantly, this limitation is often concealed by the improved performance on tasks that do not require understanding the visual context. As a result, how to effectively enhance MICL ability and reliably evaluate the MICL performance remains underexplored. To address these issues, we first introduce **D**ynamic **A**ttention **ReA**llocation (**DARA**), an efficient fine-tuning strategy that encourages models to attend to the visual context by rebalancing attention across visual and textual tokens. In addition, we present **TrueMICL**, an MICL-dedicated dataset with both support and test sets that explicitly requires the integration of multimodal information—particularly visual content—for correct task completion. Extensive experiments demonstrate the effectiveness of our holistic solution, showcasing substantial improvements in the true multimodal in-context learning capabilities. Code and datasets are available at here.

## 1 Introduction

Multimodal Large Language Models (MLLMs) have extended the emergence of in-context learning (Brown et al., 2020; Dong et al., 2022) from Large Language Models to multimodal domains and enabled Multimodal In-Context Learning (MICL) (Alayrac et al., 2022; Jiang et al., 2024). These pre-trained models can rapidly adapt to vision-language (VL) tasks, given few-shot multimodal demonstrations (demos) consisting of images, questions, and answers, without heavy parameter adaptations (Ferber et al., 2024), as shown in Figure 1. Compared with zero-shot evaluation, MICL has shown noticeable improvement on standard VL tasks such as image captioning (Alayrac et al., 2022; Awadalla et al., 2023; Laurençon et al., 2024b).

However, various studies have identified a key limitation of current MLLMs: they struggle to effectively utilize visual information in the demonstrations (Baldassini et al., 2024; Chen et al., 2023b; Jia et al., 2024; Zong et al., 2024). Concretely, they tend to overlook visual context in the multimodal demonstrations and over-rely on textual patterns in the context,

---

\* Equal contribution.
† Correspondence: jindong.gu@outlook.com, chenshuo.cs@outlook.com

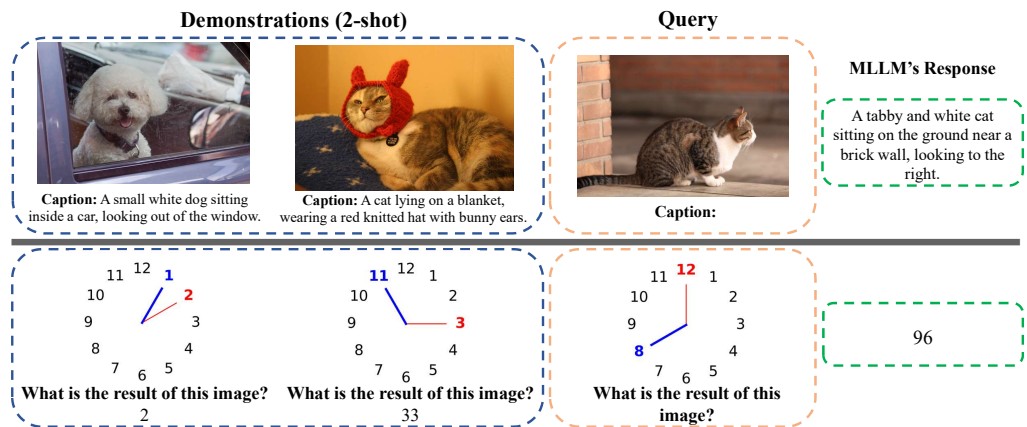

Figure 1: Examples of using MICL to solve image captioning from MSCOCO (Chen et al., 2015) (top) and Clock Math from our proposed dataset, TrueMICL (bottom). Generating captions relies more on the ability of task recognition, and a correct caption can be answered based on the text in the demos (*e.g.*, mimicking the caption style in the demos), without a deep understanding of the demo images. However, our task requires task learning where the model needs to learn the relationship between the text and images in the demos (*i.e.*, multiplying the two numbers pointed by the clock arrows) to correctly respond to the query.

which results in textual pattern imitation rather than true multimodal adaptation. This behavior renders multimodal ICL still unimodal and largely limits both its applications and practical utility. Moreover, this disadvantage is hard to discern as it can be concealed by the improved performance on tasks where reasonable responses can be generated from text pattern following, without a deep understanding of visual context. For example, Chen et al. (2023b) found that omitting the demo images during image captioning still yields comparable performance, indicating that models can generate reasonable captions for the query image even without referencing the demo images. Hence, these datasets are not suitable for evaluating MICL. To address these challenges, this study explores the following two essential research questions: *1) How to efficiently alleviate the overlooking of visual modality to truly advance MICL? 2) What kind of datasets are more suitable for improving and evaluating the true MICL ability, especially the ability to understand the visual information in the demos?*

For the first question, we introduce **D**ynamic **A**ttention **R**e**A**llocation (**DARA**) to mitigate visual context neglect and reduce overreliance on textual modality. DARA introduces a set of learnable attention-balancing parameters that dynamically regulate the influence of visual and textual tokens in demos during attention computation. DARA multiplies the columns of the attention score matrices corresponding to visual tokens by these learned parameters, thereby encouraging the model to emphasize the visual information in the demos. More importantly, DARA is lightweight, introducing only a small number of learnable parameters for rapid adaptation. Our experiments show that with around 100 parameters, DARA can yield up to a 10% improvement on downstream tasks. Consequently, DARA preserves the advantages of in-context learning while still achieving substantial performance gains.

For the second question, we propose that MICL datasets should prompt models to leverage visual context, quickly infer intent, and adapt to queries, rather than merely replicating textual answer styles or learning textual label spaces. We are motivated by the ICL disentanglement framework (Pan et al., 2023), which disentangles ICL into two components: task recognition—identifying a known task from the demonstrations and applying pre-trained priors—and task learning—acquiring new input-label mappings from the demos. Existing MICL datasets focus primarily on task recognition, requiring models to identify pre-learned tasks such as image captioning. However, they largely overlook task learning, where models learn an unseen mapping relationship between images and texts from the demos (Zong et al., 2024; Jia et al., 2024; Wang et al., 2024a). As shown in the bottom of Figure 1, the correct response needs the learn the task strategy of identifying numbers and doing the math calculation. To address this problem, we introduce **TrueMICL**, a MICL-dedicated dataset

designed with a critical principle: *correct responses must rely on a comprehensive understanding of the multimodal context, especially the visual information*. In other words, the dataset should focus more on multimodal task learning, and *a correct response must rely on the presence and understanding of demo images*. Besides, TrueMICL is designed to be scalable and configurable for different levels of difficulty and number of samples, making it practical and flexible for MICL evaluation. Specifically, we generate a dataset with both support and evaluation splits with 860 samples in total, comprising 4 types and 7 distinct tasks, covering mathematical reasoning, pattern finding, and novel visual concept learning etc.

Through extensive experiments across a range of MLLMs and tasks, we demonstrate the effectiveness of both DARA and TrueMICL. Our results show that current MLLMs find the evaluation tasks from TrueMICL quite challenging, and DARA significantly improves the MICL performance on both our evaluation tasks and standard VL tasks. This highlights the potential of DARA to advance MICL ability and the importance of TrueMICL for more suitable MICL evaluation. Our contributions can be summarized as follows:

- To enhance the MICL ability and especially to cure the visual context neglect, we propose DARA, an effective and efficient fine-tuning approach, to boost the influence of visual modality in the context. With only a few additional learnable parameters, DARA achieves substantial MICL performance improvements.
- Towards more reliable MICL improvement and evaluation, we curate TrueMICL, a MICL-specialized dataset generation pipeline focused on emphasizing the critical role of visual information in multimodal contexts to improve MICL performance.
- Comprehensive experiments across diverse MLLMs show the effectiveness of both DARA and TrueMICL, providing valuable empirical insights into the effective strategies for true multimodal in-context learning ability.

## 2 Related Work

**Multimodal In-Context Learning.** Built on powerful language backbones, some MLLMs start to show MICL ability, such as Flamingo (Alayrac et al., 2022), Qwen2-VL (Wang et al., 2024b), Idefics3 (Laurençon et al., 2024a), and Phi-3.5-Vision (Abdin et al., 2024), etc. These models support interleaved multiple image-text pairs as input and can generate responses to the query image-question pair conditioned on the previous image-text demonstrations. Another line of work focuses on understanding the working mechanism of MICL (Chen et al., 2023b; Li et al., 2023c; Yang et al., 2024b; Baldassini et al., 2024; Qin et al., 2024; Luo et al., 2024; Xu et al., 2024; Zhang et al., 2024; Li et al., 2025; Huang et al., 2025). Some focus on exploring better demonstration configurations for VQA (Li et al., 2023c) and Image Captioning (Yang et al., 2024b). Some (Baldassini et al., 2024; Luo et al., 2024; Chen et al., 2023b) examine the essential factor that contributes to MICL and also find that current MICL relies primarily on text-driven mechanisms, showing little to no influence of the visual modality. Zong et al. (2024) points out the limitations of using conventional vision-language tasks to evaluate MICL. Their benchmark indicates that most models face substantial challenges performing MICL tasks without providing alleviation methods. Given such limited performance, we propose a holistic solution to advance the MICL ability by encouraging the model to see the demo images with the help of our fine-tuning method DARA and dataset TrueMICL.

**Fine-tuning MLLMs to enhance MICL ability.** Some studies (Zhao et al., 2023; Doveh et al., 2024; Li et al., 2023b;a; Gao et al., 2025; Li et al., 2023b) have tried fine-tuning the MLLMs for better MCIL ability mainly by adjusting the dataset format. For example, Zhao et al. (2023) transforms interleaved data into a unified context and Doveh et al. (2024) extends the instruction tuning dataset into the form of multi-image conversation. However, the models still suffer from over-dependency on the text modality, and the evaluation datasets are mainly traditional VL tasks that can be answered based solely on the query images. Jia et al. (2024) replaces the original text answers in the demos with semantically irrelevant strings to encourage the model to focus more on the images. However, the proposed method requires heavy direct reference optimization. Different from these, our work proposes a holistic solution consisting of both a lightweight fine-tuning method, DARA, and a dedicated MICL dataset, TrueMICL, to improve and evaluate the MICL ability.

# 3 Methodology

We begin by briefly formulating MICL and introducing DARA in Section 3.1, followed by the introduction of TrueMICL in Section 3.2.

## 3.1 Dynamic Attention Reallocation

In multimodal in-context learning, an input query $q$, consisting of an image $I_q$ and a question or instruction $T_q$, follows a context prompt $C_q$ are passed to a pre-trained MLLM $f$. The context prompt $C_q$ includes $N$ task demonstrations drawn from a support set $S$, where each demonstration comprises an image $I_i$, instruction $T_i$, and response $R_i$. The MLLM $f$ generates a response $R_q$ to the query $q$—for instance, answering $T_q$—based on both the query components $I_q$ and $T_q$, and the multimodal context $C_q$. Formally, the process can be represented as: $R_q = f([C_q, q])$, where $q = \langle I_q, T_q \rangle$, $C_q = \{\langle I_i, T_i, R_i \rangle\}_N$.

Previous studies (Chen et al., 2023b; Baldassini et al., 2024; Luo et al., 2024) have identified the limited influence of context images $\{I_i\}_N$ on MICL performance. Chen et al. (2023b) analyzed attention patterns and suggested that this lack of sensitivity to visual information in the context may stem from the model's imbalanced attention allocation between images and texts. To address this imbalance, we propose Dynamic Attention Reallocation (DARA), which effectively amplifies the influence of visual tokens within the attention mechanism.

Considering one decoder layer in the language backbone of an MLLM, it includes the crucial self-attention module (Vaswani et al., 2017), which maps input hidden representations $\mathbf{X}$ into queries $\mathbf{Q}$, keys $\mathbf{K}$, and values $\mathbf{V}$ via linear projections using weights $\mathbf{W}_Q, \mathbf{W}_K, \mathbf{W}_V$. The decoder layer's output $\mathbf{O}$ is then calculated as $\mathbf{O} = \text{softmax}(\mathbf{S})\mathbf{V}$, where $\mathbf{S} = \mathbf{Q}\mathbf{K}^T$ represents the attention score matrix. The $j$th column $\mathbf{S}_{\cdot j}$ in the attention score matrix $\mathbf{S}$ indicates the relative influence of the $j$th token over other tokens in the input sequence. To encourage the model to focus more on context images, we directly introduce a set of learnable parameters to dynamically adjust the attention scores for columns corresponding to images. Specifically, the attention scores for visual tokens from context images are scaled by these learnable parameters. We implement this by introducing an attention balancer factor $\mathbf{F}$ to the softmax operation, represented as $\mathbf{S}' = \mathbf{S}\mathbf{F}$, where $\mathbf{F} = \text{diag}(\mathbf{f}) \in \mathbb{R}^{l \times l}$ and $\mathbf{f} \in \mathbb{R}^l$ is a vector of learnable parameters. Only the positions in $\mathbf{f}$ corresponding to visual tokens from $I$ are non-zero, while other positions remain zero. The modified attention output is then computed as $\mathbf{O}' = \text{softmax}(\mathbf{S}')\mathbf{V}$. Parameters in $\mathbf{F}$ can be trained by normal cross-entropy loss and all the other parameters in MLLMs are frozen.

---

**Algorithm 1** Pseudocode of DARA.

scale: *divide by the square root of dimension.*

```
1 # Compute attention scores and apply amplification
2 scores = scale(torch.matmul(query, key.T))
3 # Adjust the attention scores of each input image
4 for img_pos_ind, img_ind in image_indexes:
5     # multiply learned weights for each image with the
              corresponding columns
6     scores[:, img_pos_index] *= DARA_weight[img_ind]
7 # Final attention scores and output
8 scores = softmax(scores + mask)
9 attention_output = torch.matmul(scores, value)
```

---

DARA is simple to implement, with PyTorch-style pseudocode provided in Algorithm 1 on the left. After calculating the attention score matrix (Line 2) in one of the attention heads, each score column corresponding to the input images $I$ is scaled by the learned weights for those images (Lines 4–6). The adjusted score matrix is then used to compute the final attention output of the modified attention head. Different heads can have different DARA modules for more flexible adaptation.

**Relationship between DARA and Low-Rank Adaptation.** Low-Rank Adaptation (LoRA) (Hu et al., 2021; Mao et al., 2025; Yang et al., 2024a; Liu et al., 2024b; Gu et al., 2023; Chen et al., 2023a) is a popular approach for parameter-efficient fine-tuning, which utilizes low-rank decomposition matrices to update parameters. Appendix A mathematically proves that, for any $\mathbf{f}$, an equivalent low-rank decomposed update can achieve the same output. In other words, DARA can be interpreted as a concise version of LoRA for MICL with many fewer trainable parameters. Experimental in Appendix D.1 also show that, given the same scale of trainable parameters, DARA can achieve better results compared to LoRA. Besides, DARA can be applied to LoRA to improve MICL performance further.

## 3.2 TrueMICL, a MICL-dedicated Dataset

To design a dataset with indispensable visual context, we are motivated by the framework from Pan et al. (2023), which disentangles in-context learning (ICL) ability into two components: *task recognition* and *task learning*. Task recognition involves identifying an already known task from demonstrations and applying pre-trained priors, such as performing general visual question answering or image captioning. In contrast, task learning requires learning a new input-label mapping from demonstrations. Task learning is scale-dependent and exhibits significant discrepancies across models of varying sizes (Pan et al., 2023).

Several studies (Zhao et al., 2023; Doveh et al., 2024; Li et al., 2023b;a; Liu et al., 2024a) have proposed evaluation and fine-tuning datasets for MICL. Examples include collecting more data in unified formats (Zhao et al., 2023) and converting data into multi-image conversations (Doveh et al., 2024). However, these datasets do not sufficiently encourage models to acquire new mapping relationships from demonstrations. Instead, they focus on tasks similar to pre-training or instruction tuning datasets, such as image captioning and general visual question answering. This leads to a critical limitation: *query questions can often be answered by either inspecting only the query image or imitating the answer style in the demo text, without utilizing the demonstration images*. This approach falls short of achieving the true multimodal in-context learning, as it neglects the core task learning ability. In this setup, no new image-text pair knowledge is required to answer the query question. Instead, the model merely needs to recognize the task and align the response with the desired text label space. As a result, fine-tuning on such datasets can make the model tend to ignore the demo images, as it is unnecessary for the correct final response to the query. More importantly, without a proper evaluation dataset, this limitation is hard to detect. Because such an issue can be concealed by the improved performance on evaluation datasets, where the performance is mainly from textual logic imitation (*i.e.*, task recognition) rather than truly understanding the novel multimodal context (*i.e.*, task learning).

To comprehensively evaluate and incentivize MICL ability, besides evaluating task recognition, another core but usually missing principle is that *correct responses must depend on an accurate understanding of the multimodal context, particularly the visual information*. In other words, the demos should present image-text pairs with relations that are less unknown to the model so that the model can conduct task learning given the context. Additionally, the task should become unsolvable if the context images are removed. To this end, we have designed a novel MICL-dedicated dataset, TrueMICL, guided by the following principles: 1) *Context dependency*: The task must be unsolvable without the context images. 2) *Novelty*: The task should introduce novel image-text relationships that are uncommon in pre-training or instruction tuning, to effectively challenge task learning ability. 3) *Perceivable Visual Information*: The necessary information extracted from the images should not be overly complex, ensuring that the visual encoder can perceive it accurately. This allows us to focus on MICL ability rather than visual perception challenges. 4) *Compatibility with Backbone*: The task should push the boundaries of multimodal in-context learning without exceeding the language backbone's capabilities. 5) *Configurability and Extensibility*: This pipeline should be easily configured to generate more data samples with different levels of difficulty. These guidelines ensure that TrueMICL effectively evaluates and enhances the model's ability to learn new tasks from multimodal contexts. In the end, we have curated 867 samples in total spanning 4 different categories, consisting of 7 distinct tasks as listed in Table 1 and more detailed information is present in Table 4 and Appendix B

## 4 Experiments

### 4.1 Experimental Setup

**MLLMs.** We utilize 3 popular open-source MLLMs capable of MICL with varying sizes, architectures and pre-training datasets: Qwen2-VL (Wang et al., 2024b), Idefics3 (Laurençon et al., 2024a), Phi-3.5-Vision (Abdin et al., 2024). We also report the performance on GPT-4o (OpenAI et al., 2024) to showcase how current closed-source models perform on our datasets. More detailed information about MLLMs is in Appendix C.1.

| Task Type | Task Full Name | Task Short Name | # Support | # Test |
|---|---|---|---|---|
| *Math Reasoning* | Operator Induction | Operator | 30 | 100 |
| | Clock Math | Clock | 30 | 100 |
| *Concept Binding* | Outlier Detection | Outlier | 30 | 100 |
| | CLEVR Count | CLEVR | 30 | 100 |
| *Pattern Finding* | Sudoku | Sudoku | 30 | 100 |
| | Palindrome Number | Palindrome | 30 | 100 |
| *Novel Concept* | Character Classification | Character | 30 | 50 |

Table 1: TrueMICL contains 4 different categories, including a total of 7 distinct tasks. More detailed information and examples are shown in Table 4 and Appendix B

**Datasets and Evaluation Metrics.** The evaluation datasets include the test split from TrueMICL and standard VL datasets. For TrueMICL, demos for each query are selected from the support set. Standard VL datasets used in our experiments include VQAv2 (Antol et al., 2015), GQA (Hudson & Manning, 2019) and A-OKVQA (Marino et al., 2019), and MSCOCO (Chen et al., 2015). The demos for query samples from these datasets are from their corresponding training split. The evaluation metric for TrueMICL is accuracy. For numerical answers, only strict matching is considered correct, while textual answers are evaluated through keyword-based matching. As for standard VL datasets, we follow the default metric for each dataset, such as accuracy for VQAv2. A more detailed introduction on datasets and metrics used in this study is present in Appendix C.2.

**Baselines.** We have conducted experiments over the following 5 baselines: *1) Zero-shot*: The model receives only the query image and text prompt, without context. *2) No-image*: The model receives the query image and text prompt, with randomly selected 4-shot demos including only the text information. *3) Random*: The 4-shot demos are randomly selected, including both images and text. *4) RICES*: The demos are selected by Retrieval-based In-Context Example Selection (Alayrac et al., 2022; Yang et al., 2022; Zebaze et al., 2024; Wang et al., 2023). For a given query, we chose the 4 most similar images from the support set and used the image-label pairs as demos. *5) LoRA*: We also fine-tune LoRA on the support set for each task while controlling the number of learnable parameters to the same scale as DARA. For more detailed settings, please refer to Appendix C.3.

**DARA.** We insert the DARA module into the first transformer layer of the language backbone in the MLLM, specifically, attention score matrices in all attention heads in the first layer will be regulated by DARA's learnable parameters. The number of learnable parameters is decided by the number of demo images and the number of heads to modify. For instance, given 5 images including 4-shot and the query image, and 32 attention heads to adjust, the total number of learnable parameters is $5 \times 32 = 160$. These learnable parameters are trained using Adam on the small support set with a learning rate of 0.001. More detailed hyperparameters are present in Table 6 in the Appendix.

## 4.2 Result Analysis on TrueMICL

The main experimental results of the models' performance on TrueMICL are shown in Table 2. We report the accuracy of three models across various tasks in TrueMICL, where each row represents a different inference method. Except for the Zero-shot, all other methods use a 4-shot setting. Additionally, the methods labeled LoRA and DARA represent the performance of models trained under the finetuning settings described in Section 4.1.

Table 2 first shows that these MLLMs perform poorly on TrueMICL in zero-shot scenarios and when demonstrations contain only text without images. Some models even achieve accuracy under 10% on certain tasks, such as Phi-3.5-Vision on Operator Induction. With randomly selected 4-shot demonstrations, the models demonstrate limited improvement in accuracy across tasks. However, applying the RICE method for selecting more relevant demonstrations yields minimal additional performance gains. This result highlights two key observations: 1) Tasks in TrueMICL truly require both visual and textual information from the context to be solved correctly; 2) Due to the model's limited ability to effectively make use of the visual information in demonstrations, simply improving the relevance of the shots (as done in RICE) is insufficient to enhance model's performance further.

| Model | Method | Operator | Clock | Outlier | CLEVR | Sudoku | Palindrome | Character | Average |
|---|---|---|---|---|---|---|---|---|---|
| **Qwen2-VL** | Zero-Shot | $13.67_{\pm0.57}$ | $24.00_{\pm2.65}$ | $41.33_{\pm2.52}$ | $34.33_{\pm1.53}$ | $88.00_{\pm1.73}$ | $30.33_{\pm1.15}$ | $23.00_{\pm3.61}$ | 36.38 |
| | No-image | $18.67_{\pm0.58}$ | $27.67_{\pm1.53}$ | $46.33_{\pm2.08}$ | $41.67_{\pm3.06}$ | $90.00_{\pm1.73}$ | $38.33_{\pm3.51}$ | $24.33_{\pm1.15}$ | 41.00 |
| | Random | $67.33_{\pm0.58}$ | $31.00_{\pm1.00}$ | $86.67_{\pm0.58}$ | $86.00_{\pm0.00}$ | $\underline{93.33}_{\pm2.08}$ | $96.00_{\pm1.00}$ | $83.33_{\pm1.15}$ | 77.67 |
| | RICES | $\underline{67.67}_{\pm0.58}$ | $31.33_{\pm1.53}$ | $\underline{87.33}_{\pm1.53}$ | $87.33_{\pm1.15}$ | $\mathbf{95.33}_{\pm1.53}$ | $95.00_{\pm1.00}$ | $\underline{94.00}_{\pm2.00}$ | 79.71 |
| | LoRA | $66.33_{\pm0.58}$ | $\underline{32.67}_{\pm0.58}$ | $\underline{87.33}_{\pm0.58}$ | $88.00_{\pm1.73}$ | $92.67_{\pm0.58}$ | $\mathbf{98.67}_{\pm1.53}$ | $\mathbf{96.00}_{\pm1.53}$ | 80.24 |
| | DARA | $\mathbf{72.67}_{\pm1.15}$ | $\mathbf{37.33}_{\pm1.53}$ | $\mathbf{91.67}_{\pm1.53}$ | $\mathbf{90.00}_{\pm1.73}$ | $\mathbf{95.33}_{\pm0.58}$ | $\underline{98.00}_{\pm1.00}$ | $\mathbf{96.00}_{\pm0.58}$ | **83.00** |
| **Idefics3** | Zero-Shot | $6.67_{\pm1.53}$ | $0.67_{\pm0.58}$ | $50.00_{\pm1.00}$ | $19.00_{\pm2.00}$ | $53.33_{\pm2.08}$ | $28.33_{\pm1.15}$ | $23.67_{\pm1.53}$ | 25.95 |
| | No-image | $10.33_{\pm2.08}$ | $2.00_{\pm1.00}$ | $52.67_{\pm1.53}$ | $23.00_{\pm1.00}$ | $59.33_{\pm2.08}$ | $28.67_{\pm2.52}$ | $27.33_{\pm3.21}$ | 29.05 |
| | Random | $16.00_{\pm1.73}$ | $9.33_{\pm0.58}$ | $\underline{59.67}_{\pm0.58}$ | $25.00_{\pm1.73}$ | $86.67_{\pm1.53}$ | $29.33_{\pm0.58}$ | $67.67_{\pm2.08}$ | 41.95 |
| | RICES | $17.33_{\pm1.53}$ | $\underline{9.67}_{\pm1.53}$ | $59.00_{\pm0.00}$ | $24.67_{\pm1.53}$ | $87.00_{\pm1.00}$ | $\underline{29.33}_{\pm0.58}$ | $77.00_{\pm1.00}$ | 43.43 |
| | LoRA | $\underline{18.67}_{\pm0.58}$ | $\underline{9.67}_{\pm1.53}$ | $59.67_{\pm0.58}$ | $24.33_{\pm0.58}$ | $\underline{87.67}_{\pm0.58}$ | $28.33_{\pm1.15}$ | $\underline{89.33}_{\pm1.15}$ | 45.38 |
| | DARA | $\mathbf{21.33}_{\pm1.53}$ | $\mathbf{14.67}_{\pm1.53}$ | $\mathbf{64.00}_{\pm2.00}$ | $\mathbf{25.00}_{\pm1.00}$ | $\mathbf{91.33}_{\pm1.15}$ | $\mathbf{33.67}_{\pm3.21}$ | $\mathbf{90.67}_{\pm1.53}$ | **48.67** |
| **Phi-3.5-vision** | Zero-Shot | $9.33_{\pm1.53}$ | $3.67_{\pm1.53}$ | $39.00_{\pm2.00}$ | $8.33_{\pm1.53}$ | $46.67_{\pm1.15}$ | $32.67_{\pm0.58}$ | $21.33_{\pm1.73}$ | 23.00 |
| | No-image | $11.33_{\pm0.58}$ | $6.33_{\pm1.53}$ | $45.33_{\pm1.15}$ | $13.00_{\pm2.00}$ | $52.67_{\pm1.15}$ | $38.00_{\pm1.73}$ | $28.33_{\pm0.58}$ | 27.86 |
| | Random | $14.33_{\pm0.58}$ | $17.33_{\pm0.58}$ | $61.67_{\pm2.08}$ | $24.33_{\pm1.15}$ | $\underline{84.67}_{\pm2.08}$ | $42.00_{\pm1.00}$ | $68.00_{\pm1.53}$ | 44.62 |
| | RICES | $\underline{15.00}_{\pm1.00}$ | $17.67_{\pm0.58}$ | $61.33_{\pm0.58}$ | $24.33_{\pm0.58}$ | $84.00_{\pm1.00}$ | $\underline{42.33}_{\pm0.58}$ | $84.67_{\pm2.00}$ | 47.05 |
| | LoRA | $14.00_{\pm0.00}$ | $\underline{16.67}_{\pm0.58}$ | $\underline{65.33}_{\pm1.15}$ | $\underline{27.67}_{\pm1.15}$ | $84.33_{\pm0.58}$ | $41.33_{\pm0.58}$ | $\underline{91.33}_{\pm2.08}$ | 48.67 |
| | DARA | $\mathbf{17.33}_{\pm1.53}$ | $\mathbf{20.00}_{\pm1.73}$ | $\mathbf{70.67}_{\pm1.53}$ | $\mathbf{32.00}_{\pm1.73}$ | $\mathbf{89.33}_{\pm1.53}$ | $\mathbf{47.33}_{\pm2.08}$ | $\mathbf{93.00}_{\pm1.00}$ | **52.81** |

Table 2: Performance of MICL from 3 different MLLMs using different methods on TrueMICL. Each column demonstrates the performance from each task in TrueMICL, with the task abbreviation as the column title. The best performance for each setting is in bold, and the second-best is underlined. DARA (rows in light red) achieves the best performance in most scenarios and outperforms baselines by a large margin. Experiments are averaged over 3 different seeds, and the performance is in percentage with the standard deviations.

However, as shown in the last two rows for each model, the performance on TrueMICL significantly improves after finetuning, particularly with the use of DARA. Compared to LoRA, which introduces several thousand parameters but yields limited gains and remains almost the same performance as random 4-shot, DARA leads to further improvements across nearly all tasks. For tasks involving math reasoning and concept binding, each model achieves an average improvement of 3 to 5 percent per task when using DARA. Meanwhile, in tasks such as pattern finding and concept learning, models like Qwen2-VL—which already achieve good accuracy using Random baseline due to their strong capabilities—still show an additional stable gain in accuracy when using DARA. Besides, the performance of the other two models on these tasks also demonstrates a stable and noticeable gain when DARA is applied, further supporting its effectiveness.

Overall, the results demonstrate that the proposed dataset, TrueMICL, serves as a reliable benchmark for evaluating true MICL capabilities, as every task category requires the model to combine both visual and textual information to provide the correct answer. Furthermore, the proposed method, DARA, consistently achieves the highest average performance across nearly all TrueMICL tasks for each model tested. This highlights the effectiveness of DARA in enhancing the model's ability to perform multimodal in-context learning.

## 4.3 Visualization of the DARA's Reallocation Effects

To better understand how DARA affects attention, we present both qualitative and quantitative visualizations as shown in Figure 2 and 3. The spatial attention heatmap (Zhang et al., 2025) of input images (Figure 2(a)) shows that, without DARA, both demonstration and query images do not receive much attention, as indicated by predominantly blue regions in the top row. After tuning with DARA, attention over image tokens increases markedly (more red/yellow areas in the bottom row), indicating enhanced attention to the visual input. While not all focus aligns with the specific object regions, the overall shift indicates that DARA encourages greater incorporation of image information.

We also quantitatively compare the attention allocation ratio over different modality tokens with and without using DARA. Figure 2(b) highlights the shift of attention on these two modalities. Without DARA, the model allocates only 28% of attention to image tokens, focusing primarily on text. With DARA, this rises to 46.7%, indicating a substantial shift toward visual content during response generation.

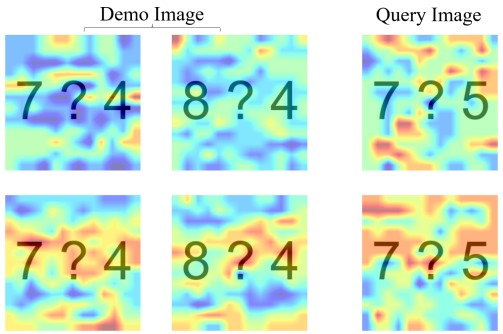 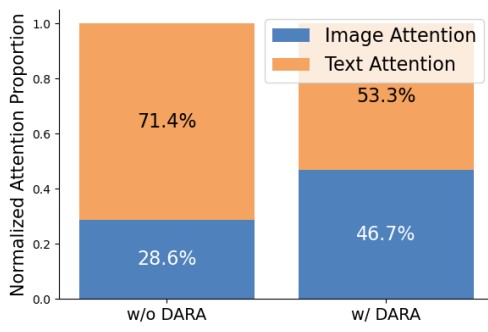

(a) Attention heatmaps over input images without (top) and with (bottom) applying DARA.

(b) Normalized attention ratios over image and text tokens without and with applying DARA.

Figure 2: DARA enhances visual attention both qualitatively (left) and quantitatively (right).

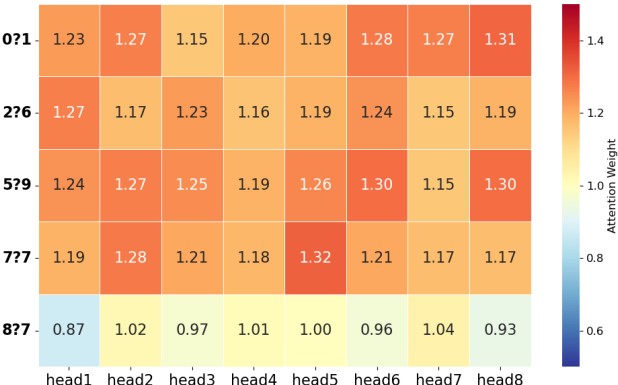

Figure 3: Learned attention amplification factors across 8 heads and 5 images in the 1st layer of Qwen2-VL. DARA introduces structured visual emphasis on demonstration images.

To further understand the value distribution of learned factors, Figure 3 visualizes these factors over four demos and one query image from the first transformer layer (8 heads) of Qwen2-VL. Values larger than 1 indicate attention amplification on visual tokens. DARA induces a clear redistribution: (a) demo images consistently receive factors larger than 1, encouraging stronger reliance on context; and (b) different heads specialize—for example, Head 1 emphasizes Demo 2 (1.27), while Head 5 emphasizes Demo 4 (1.32). These patterns confirm that DARA enables selective, context-aware visual attention during MICL.

## 4.4 Result Analysis on Standard VL Datasets

We also analyzed the performance of DARA on standard VL datasets, following the same baseline configuration as described in Section 4.1. The results are summarized in Table 8 in the Appendix. The results show that, for standard VL tasks, model performance relies primarily on the model's own ability to understand the query image and associated textual information. By comparing the first three settings, no demonstrations (Zero-shot), demonstrations without images (No-image), and random 4-shots, we observe almost no differences in performance. Taking Qwen2-VL on VQAv2 as an example, the accuracy under these settings is 78.6%, 78.7%, and 79.1%, respectively. These results align with previous findings (Chen et al., 2023b) and indicate that correct responses for these standard vision tasks do not depend on the visual context provided in the demonstrations. Once the model has extracted sufficient information from the query image and integrated it with the textual context, a high-quality response can already be generated.

Moreover, by examining the performance of DARA, we observe that DARA—designed to encourage the model to better focus on visual content in the demonstrations—achieves results comparable to the 0-shot and 4-shot settings. This finding confirms, on the one hand, that standard VL benchmarks are indeed suboptimal for evaluating true multimodal in-context learning, as they do not even require visual context; and on the other hand, that the use of DARA does not lead to performance degradations on these standard VL tasks.

| Qwen2-VL | Operator | Clock | Outlier | CLEVR | Sudoku | Palindrome |
|---|---|---|---|---|---|---|
| **Baseline** | 67.33% | 31.00% | 86.67% | 86.00% | 93.33% | 96.00% |
| **Operator** | – | 31.67% (+0.67%) | 90.33% (+3.66%) | 89.33% (+3.33%) | 95.67% (+2.34%) | 97.00% (+1.00%) |
| **Clock** | 72.00% (+4.67%) | – | 88.00% (+1.33%) | 88.67% (+2.67%) | 95.00% (+1.67%) | 96.00% (+0.00%) |
| **Outlier** | 68.33% (+1.00%) | 32.00% (+1.00%) | – | 87.67% (+1.67%) | 94.00% (+0.67%) | 95.33% (+1.33%) |
| **CLEVR** | 67.00% (-0.33%) | 31.00% (+0.00%) | 87.67% (+1.00%) | – | 95.33% (+2.00%) | 97.00% (+1.00%) |
| **Sudoku** | 72.00% (+4.67%) | 31.33% (+0.33%) | 89.33% (+2.66%) | 89.00% (+3.00%) | – | 95.67% (-0.23%) |
| **Palindrome** | 71.00% (+3.67%) | 31.67% (+0.67%) | 89.67% (+3.00%) | 89.67% (+3.67%) | 97.00% (+3.67%) | – |

Table 3: Transferability performance of DARA across tasks. Diagonal cells (self-transfer) are omitted for clarity. Each column reports the performance when DARA is trained on other tasks and evaluated on the column task. DARA shows good transferability and consistently achieves transfer gains in most settings, with improvements up to 4.67%.

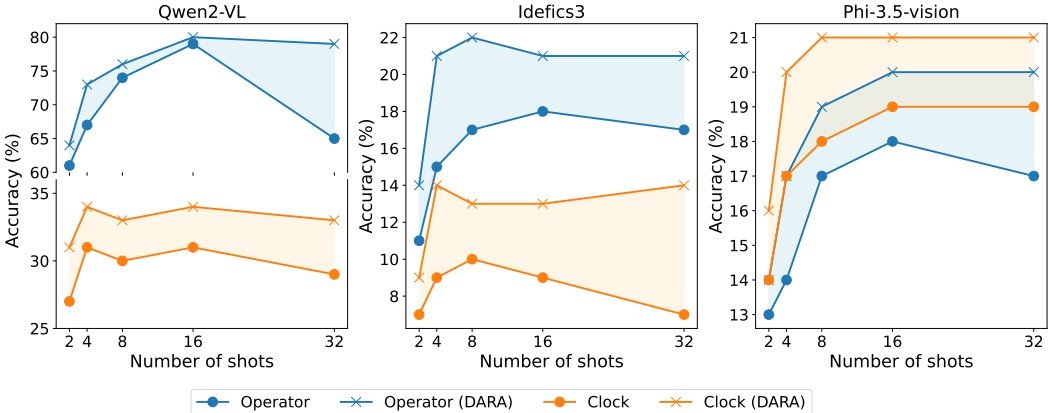

Figure 4: Performance comparison on Operator and Clock tasks across three models with different numbers of shots, ranging from 2 to 32. Compared to the baseline Random method, DARA consistently improves the MICL performance across different numbers of shots.

## 4.5 Ablation Study

We further conduct various ablation studies on DARA to verify the effectiveness of our design. A more detailed discussion is in Appendix D

**Transferability on untuned tasks.** We evaluate the transferability of DARA by testing whether improvements on one task can transfer to untrained ones. As shown in Table 3, training on a single task yields 2–5% higher accuracy on unseen tasks compared to standard 4-shot baseline inference. While this does not reach the performance of models trained directly on each target task, it demonstrates that DARA can introduce transferability, enabling better generalization across related MICL tasks.

**Impact of Increasing Shots.** We also evaluated the performance of models fine-tuned with TrueMICL under different shot settings. As shown in Figure 4, we present a comparison between the performance of models trained with varying numbers of shots and the corresponding baseline at that shot level. Within a certain range of shot numbers (for example, up to 10 shots for Qwen2-VL, determined by the maximum number of images the model can reliably process at once), applying DARA consistently brings performance improvements across all shot settings. However, when the number of shots becomes too large, neither DARA nor LoRA can consistently guarantee further gains. This finding suggests that as more capable models emerge in the future, DARA will still prove to be effective.

**Prompt Design.** To test whether DARA's improvements originate from prompt engineering, we compare our original minimal prompt—which avoids leaking visual cues through text—with an enhanced version that includes detailed instructional cues (Sahoo et al., 2024). As shown in Appendix D.4, while such prompts help standardize response formatting, they

do not lead to significant accuracy gains. This supports that DARA's effectiveness arises from improved reasoning over demonstrations, rather than prompt tuning.

**Human Evaluation.** To assess whether TrueMICL tasks truly require MICL ability, we conducted a human study involving 20 participants. In 0-shot setting, where no demos were provided, participants generally failed to answer. However, their performance improved once demonstrations were included. These results suggest that the tasks cannot be solved using prior knowledge or superficial cues alone. Instead, successful completion requires learning from the multimodal context, thereby validating the design of our benchmark.

**Layer Design on DARA.** We further investigated applying DARA to multiple transformer layers beyond only the first layer. While extending DARA to deeper layers yields comparable accuracy, it leads to increased computational and parameter overhead. We attribute this to the first layer's unique role in initiating early cross-modal fusion, as later layers process already entangled representations, limiting the impact of targeted amplification. Thus, restricting DARA to the first layer achieves a better balance between effectiveness and efficiency. The result of applying different layers is shown in Appendix D.3

**Hard-coded Attention Adjustment.** As an additional baseline, we experimented with a hard-coded attention amplification strategy. Specifically, we modified attention logits to completely ignore text tokens for half of the heads, forcing attention exclusively onto image tokens. This rigid masking led to unstable and incoherent outputs, likely due to disrupted modality balance. In contrast, DARA employs a learnable amplification factor that softly increases attention to image tokens, allowing the model to dynamically adjust during training. This preserves output fluency while improving performance.

**Evaluation on GPT-4o.** We also tested TrueMICL on GPT-4o, a state-of-the-art closed-source model. In the 0-shot setting, GPT-4o fails on most tasks except a few logic-based ones like Sudoku. With 4-shot demonstrations, it achieves significantly better accuracy (full results in Appendix D.2), highlighting the indispensable role of context images in TrueMICL. However, despite GPT-4o's strong performance, certain fundamental MICL challenges persist; for example, GPT-4o's accuracy significantly decreases when facing more challenging tasks such as our specially designed harder Sudoku variant. Furthermore, due to the closed-source nature of GPT-4o, it remains unclear whether these performance gains are driven purely by scaling or by undisclosed architectural and training strategies. Importantly, the lightweight design of DARA complements these powerful models, providing further improvements even when combined with parameter-efficient fine-tuning approaches like LoRA. Consequently, DARA remains valuable in addressing attention imbalance, especially benefiting openly accessible, resource-constrained models widely used in diverse applications. Moreover, TrueMICL serves as a robust benchmark for accurately evaluating the MICL capabilities of both open-source and proprietary MLLMs.

# 5 Conclusions

Despite recent advances in multimodal large language models, our study reveals a critical limitation in current approaches to Multimodal In-Context Learning (MICL): the tendency to overlook visual information in demonstrations and over-rely on textual patterns. This undermines the fundamental promise of MICL, reducing it to a predominantly unimodal process and limiting its real-world applicability. To tackle this issue, we propose DARA, a lightweight yet effective fine-tuning method that dynamically reallocates attention to enhance the influence of visual tokens in the in-context examples. Complementing this, we introduce TrueMICL, a challenging and diagnostic dataset that emphasizes true multimodal adaptation by requiring visual understanding for task success. Our extensive experiments across diverse MLLMs demonstrate that DARA consistently improves performance on both standard and TrueMICL tasks, validating its effectiveness in enhancing genuine multimodal adaptation. Furthermore, TrueMICL exposes existing blind spots in MLLMs and serves as a valuable benchmark for evaluating true MICL capabilities beyond textual-level imitation. Together, DARA and TrueMICL offer a holistic framework for truly advancing and evaluating MICL, paving the way for more faithful multimodal ICL in future MLLMs.

## Acknowledgment

This paper is supported by the DAAD programme Konrad Zuse Schools of Excellence in Artificial Intelligence, sponsored by the Federal Ministry of Research, Technology and Space.

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

## A  DARA as a constrained version of LoRA

Consider a transformer-based attention mechanism with query, key, and value transformations defined by $\mathbf{W}_Q, \mathbf{W}_K, \mathbf{W}_V$, and let $\mathbf{S} := \frac{\mathbf{Q}\mathbf{K}^\top}{\sqrt{d}}$ be the attention score matrix, where $\mathbf{Q} := \mathbf{X}\mathbf{W}_Q$ and $\mathbf{K} := \mathbf{X}\mathbf{W}_K$, where $\mathbf{X}$ is the input tensor to the transformer attention module. Suppose we introduce an attention reallocation factor to the softmax operation by defining $\mathbf{F} := \mathrm{diag}(\mathbf{f}) \in \mathbb{R}^{L \times L}$, where $\mathbf{f} \in \mathbb{R}^L$ is a vector of learnable factors. The modified scores are $\mathbf{S}' := \mathbf{S}\mathbf{F}$. On the other hand, we consider the update (e.g., from gradient descent algorithm) on the module weights $\mathbf{W}_Q, \mathbf{W}_K$, defined as $\Delta\mathbf{W}_Q, \Delta\mathbf{W}_K$. After updating the weights, we have $\mathbf{W}'_Q := \mathbf{W}_Q + \Delta\mathbf{W}_Q, \mathbf{W}'_K := \mathbf{W}_K + \Delta\mathbf{W}_K$, the updated $\mathbf{Q}' = \mathbf{X}\mathbf{W}'_Q, \mathbf{K}' = \mathbf{X}\mathbf{W}'_K$, and the updated attention score matrix $\mathbf{S}'' := \frac{\mathbf{Q}'\mathbf{K}'^\top}{\sqrt{d}}$. *Prove that for $\forall \mathbf{f}, \exists \Delta\mathbf{W}_Q$ and $\exists \Delta\mathbf{W}_Q$, so that $\mathbf{S}' = \mathbf{S}''$*

**Proof.** We want:

$$\mathbf{S}'' = \frac{\mathbf{Q}'\mathbf{K}'^\top}{\sqrt{d}} = \frac{\mathbf{Q}\mathbf{K}^\top}{\sqrt{d}}\mathbf{F} = \mathbf{S}\mathbf{F} \tag{1}$$

Equivalently:

$$\mathbf{Q}'\mathbf{K}'^\top = \mathbf{Q}\mathbf{K}^\top\mathbf{F}. \tag{2}$$

We need to express $\mathbf{Q}\mathbf{K}^\top\mathbf{F}$ as $\mathbf{Q}'\mathbf{K}'^\top$ with $\mathbf{Q}' = \mathbf{X}\mathbf{W}'_Q$ and $\mathbf{K}' = \mathbf{X}\mathbf{W}'_K$.

There is no unique solution. In fact, there is a family of solutions parameterized by an invertible matrix $\mathbf{R} \in \mathbb{R}^{d \times d}$. Consider:

$$\mathbf{Q}' = \mathbf{Q}\mathbf{R} \quad \text{and} \quad \mathbf{K}' = \mathbf{F}\mathbf{K}\mathbf{R}^{-T}. \tag{3}$$

Then:

$$\mathbf{Q}'\mathbf{K}'^\top = (\mathbf{Q}\mathbf{R})(\mathbf{R}^{-1}\mathbf{K}^\top\mathbf{F}) = \mathbf{Q}\mathbf{K}^\top\mathbf{F}. \tag{4}$$

Since $\mathbf{Q} = \mathbf{X}\mathbf{W}_Q$ and $\mathbf{K} = \mathbf{X}\mathbf{W}_K$, we have:

$$\mathbf{Q}' = \mathbf{X}\mathbf{W}'_Q = \mathbf{X}\mathbf{W}_Q\mathbf{R} \implies \mathbf{W}'_Q = \mathbf{W}_Q\mathbf{R}, \tag{5}$$

if $\mathbf{X}$ is full rank or using a suitable inverse/pseudoinverse.

$$\mathbf{K}' = \mathbf{X}\mathbf{W}'_K = \mathbf{F}\mathbf{X}\mathbf{W}_K\mathbf{R}^{-T} \implies \mathbf{W}'_K = \mathbf{X}^+\mathbf{F}\mathbf{X}\mathbf{W}_K\mathbf{R}^{-T}. \tag{6}$$

Here, $\mathbf{X}^+$ denotes a left-inverse or pseudoinverse of $\mathbf{X}$. Since $\mathbf{R}$ is arbitrary and invertible, there are infinitely many such solutions. Hence, for any given $\mathbf{F}$, we can always find $\Delta\mathbf{W}_Q = \mathbf{W}'_Q - \mathbf{W}_Q$ and $\Delta\mathbf{W}_K = \mathbf{W}'_K - \mathbf{W}_K$ to achieve $\mathbf{S}'' = \mathbf{S}'$.

*Prove that each corresponding* $\Delta\mathbf{W}_Q, \Delta\mathbf{W}_K$ *can be decomposed into a low-rank matrices multiplications, such as* $\Delta\mathbf{W}_Q = \mathbf{AB}, \Delta\mathbf{W}_K = \mathbf{CD}$

**Proof.** Any matrix admits a factorization into two matrices of potentially lower rank. For a given $\Delta\mathbf{W}_Q$, consider its singular value decomposition (SVD):

$$\Delta\mathbf{W}_Q = \mathbf{U}\boldsymbol{\Sigma}\mathbf{V}^\top, \tag{7}$$

where $\mathbf{U}$ and $\mathbf{V}$ are orthonormal matrices and $\boldsymbol{\Sigma}$ is diagonal with nonnegative singular values. By retaining only the nonzero singular values (or even a subset to achieve a rank reduction), we obtain a low-rank factorization:

$$\Delta\mathbf{W}_Q = \mathbf{U}_r\boldsymbol{\Sigma}_r\mathbf{V}_r^\top, \tag{8}$$

with $\mathbf{U}_r \in \mathbb{R}^{d_{\text{in}}\times r}$, $\boldsymbol{\Sigma}_r \in \mathbb{R}^{r\times r}$, and $\mathbf{V}_r^\top \in \mathbb{R}^{r\times d}$. Define $\mathbf{A} = \mathbf{U}_r\boldsymbol{\Sigma}_r \in \mathbb{R}^{d_{\text{in}}\times r}$ and $\mathbf{B} = \mathbf{V}_r^\top \in \mathbb{R}^{r\times d}$. Thus:

$$\Delta\mathbf{W}_Q = \mathbf{AB}. \tag{9}$$

A similar argument applies to $\Delta\mathbf{W}_K$. Hence:

$$\Delta\mathbf{W}_K = \mathbf{CD} \tag{10}$$

for some $\mathbf{C} \in \mathbb{R}^{d_{\text{in}}\times r'}$ and $\mathbf{D} \in \mathbb{R}^{r'\times d}$, possibly with the same or different rank $r'$.

To conclude, the updates $\Delta\mathbf{W}_Q$ and $\Delta\mathbf{W}_K$ obtained to achieve $\mathbf{S}'' = \mathbf{S}'$ can always be decomposed into products of lower-rank matrices. This establishes that these parameter updates can be implemented in a low-rank form, aligning with practical low-rank adaptation methods. It is necessary to clarify that DARA can be seen as a constrained variant of LoRA rather than a technical equivalent, as it requires input-dependent updates and token-position-specific factors that deviate from the standard LoRA assumptions. Specifically, the weight updates in LoRA are input-independent and fixed after learning, whereas the desired low-rank update $\Delta\mathbf{W}_K$ depends on the specific input $\mathbf{X}$ as shown in the Formula 6 However, such constraints actually can provide practical benefits to MICL. While LoRA is more general, it can be indirect and suboptimal to rebalance the attention allocation for better MICL. In contrast, DARA's constrained formulation enables targeted visual attention modulation with far fewer parameters, leading to more efficient MICL improvement.

## B TrueMICL

Guided by the principles in Section 3.2, we designed and compiled four categories encompassing a total of seven distinct tasks, as illustrated in Table 1 and 4. TrueMICL encompasses a wide range of MICL capabilities, including mathematical reasoning, novel concept binding, and pattern interpretation, providing a comprehensive dataset for multimodal in-context learning. Each task is designed with adjustable difficulty levels, such as more diverse concepts in novel concept binding, more complex visual patterns in pattern interpretation, etc. Each task in TrueMICL is divided into a support set containing 30 samples and a different test set containing 100 queries (except for Character Classification, due to the limited number of character images sourced from movie scenes).

**Operator Induction** is derived from Zong et al. (2024). The dataset consists of simple arithmetic operations formed by three operators: addition, subtraction, and multiplication. Since the operator used in each image is not explicitly provided, the model must analyze the relationship between the image content and the answer to respond to the query.

**Clock Math** is an original task inspired by the arithmetic pattern. The model must first perceive the number in the image, then learn the relationship between the number and the answer (*e.g.*, addition), to correctly answer the query.

For arithmetic tasks, considering the current capabilities of existing models, our dataset adopts the most straightforward form, using actual digits to perform the whole operations. In fact, replacing digits with arbitrary symbols to express implicit operations can all be seen as an extension of this type of task in higher levels of difficulty.

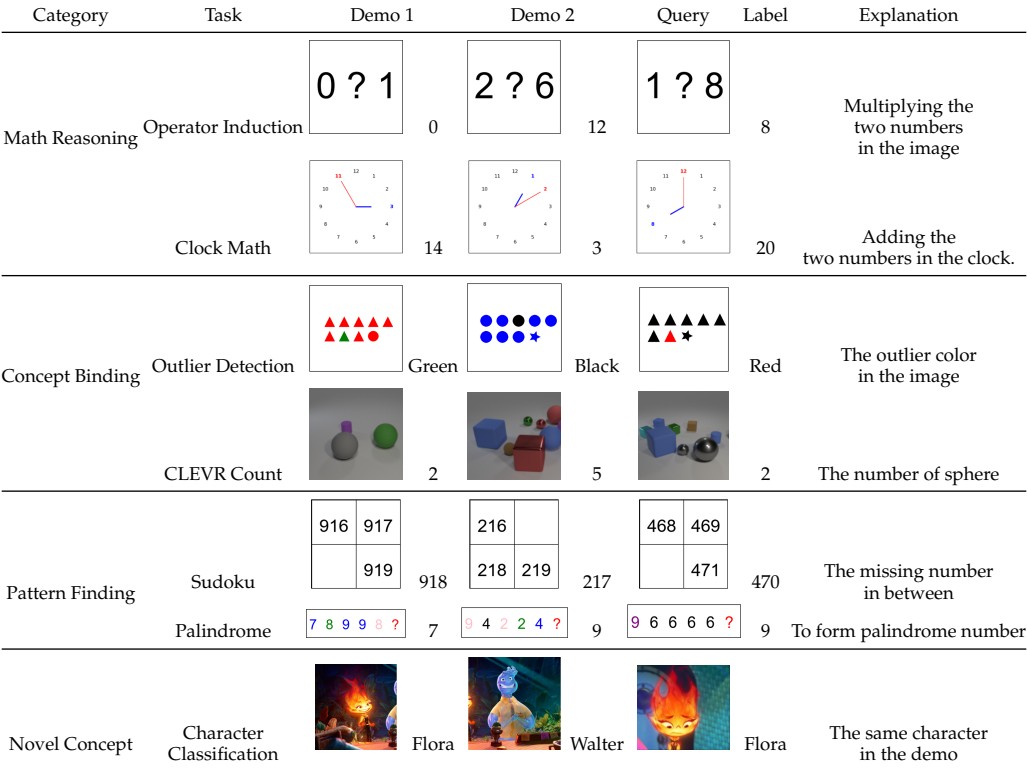

Table 4: An overview of task examples in TrueMICL. The label to the query requires the model to learn the relationship between images and text in the demos.

**Outlier Detection** is originally designed to evaluate the model's binding capability. Each image in this dataset contains varying numbers of two different shapes and colors. The model needs to identify the desired outlier feature based on the demo text.

Similarly, in concept binding tasks, features such as shape, color, and size can be replaced with less explicit attributes, such as visual appearance or texture. This substitution will make the task significantly more challenging.

**CLEVR Count** is derived from Zong et al. (2024). Each image contains objects with varying attributes such as colors and shapes. The model will be asked to count the number of objects with specific attributes based on the demonstrations. We simplified the samples to match the model's visual perception capabilities. Specifically, we removed hard-to-distinguish attributes such as material and size, kept only color and shape, and redesigned the prompts to align with our intended focus on concept binding in the dataset.

**Sudoku** and **Palindromic Number** are originally designed for pattern finding. They contain images of incomplete Sudoku grids or partially missing sequences of palindromic numbers. To solve these tasks, the model must recognize the underlying patterns behind the multimodal demos and apply this learned relation to correctly respond to queries.

For pattern finding tasks, taking Sudoku as an example, increasing the number of missing digits or introducing additional implicit rules for number placement can substantially raise the task difficulty. High-level reasoning problems commonly used in IQ or logic tests also fall into this category as more advanced tasks of pattern finding.

**Character Classification** is originally designed to test the ability of novel concept learning from multimodal demos. We collect movie character images after the model's cutoff date and assign previously unseen names. This setup requires the model to associate novel names with unfamiliar appearances through the demonstrations in order to correctly answer the queries.

For this unique type of classification task, any similarly constructed dataset that uses images collected after the release of the evaluated models can be considered valid.

# C More Details of Experimental Setup

## C.1 MLLMs

| Model | Qwen2-VL-7B-Instruct | Idefics3-8B-Llama3 | Phi-3.5-Vision-Instruct |
|---|---|---|---|
| # Params | 7B | $\sim$8.5B | 4.2B |
| Visual Encoder | Custom Patch Encoder | SigLIP-SO400M | CLIP ViT-L/14 |
| Language Model | Qwen2-7B-Instruct | LLaMA3-8B-Instruct | Phi-3.5-mini |
| Architecture | Autoregressive | Autoregressive | Autoregressive |
| # Visual Tokens | Up to 16,384 | 169 (364×364) | Dynamic (block concat) |

Table 5: 3 MLLMs used in this study, with various architectures, sizes, and number of visual tokens per image.

Table 5 shows more details about the models used in this research. The following is a brief introduction to all 3 MLLMS.

Qwen2-VL is a state-of-the-art multimodal model excelling in visual understanding. It supports multilingual OCR and dynamically adapts to arbitrary image resolutions for human-like visual processing. The link to the model on Hugging Face: Qwen2-VL-7B-Instruct.

Idefics3 is an open multimodal model that processes arbitrary sequences of images and text to generate text outputs. It supports tasks like visual question answering, image-based storytelling, and pure language modeling, with improved OCR, document understanding, and visual reasoning over previous versions. The link to the model on Hugging Face: Idefics3-8B-Llama3.

Phi-3.5-Vision is a lightweight, state-of-the-art multimodal model from the Phi-3 family, trained on high-quality text and vision data. It is enhanced via supervised fine-tuning and direct preference optimization to ensure strong instruction following and safety. The link to the model on Hugging Face: Phi-3.5-Vision-Instruct.

## C.2 Datasets and Evaluation Metrics

To construct the dataset, each task type is first divided into four subsets: the query set, the support set, the training-query set, and the training-support set. These subsets are mutually exclusive and contain no overlapping samples. Depending on whether the setting is inference or fine-tuning, we then organize the data into either an inference file or a training file accordingly, with the desired number of shots. While choosing demos, only those demos of the same type of query will be chosen. This structure ensures clear separation between evaluation and training data, and allows for flexible adaptation to different experimental settings.

To assess the performance of our model on our proposed dataset, we primarily adopt **accuracy** (in percentage) as the evaluation metric. For numerical answers, only strict matching is considered correct, while textual answers are evaluated through keyword-based matching. We additionally perform manual verification on a subset of the predictions to ensure the reliability of the scoring. The final accuracy is computed as:

$$\text{Accuracy} = \frac{N_{\text{correct}}}{N_{\text{total}}} \times 100\%$$

where $N_{\text{correct}}$ denotes the number of correctly predicted answers and $N_{\text{total}}$ represents the total number of evaluated samples.

### C.3 Baselines

To successfully apply DARA, we first initialize a new `nn.Module` inside the model, with dimensions [`number of attention layers, number of attention heads, number of images`]. Then, for each input, we extract the position of each image from the `input_ids`. By progressively passing the new parameters into the attention layer, we are able to apply a scalar factor to each image token in the attention matrices of the specified layers and heads. All of the factors are first initialized to 1. As the model is finetuned using the settings described in Table 6, this parameter matrix is updated automatically.

| Parameter | Value |
|---|---|
| Batch size per device | 1 |
| Gradient accumulation steps | 4 |
| Epochs | 5 |
| Learning rate | 1e-3 |
| Warmup steps | 5 |
| Optimizer | AdamW (8-bit) |
| Weight decay | 0.01 |
| Learning rate scheduler | Linear |
| Precision | fp16 / bf16 (auto) |
| Max sequence length | 2048 |
| Random seed | 3407 |
| Output directory | "outputs" |

Table 6: General Supervised Fine-tuning Configuration

To ensure a fair comparison in parameter scale between LoRA and DARA, we restrict LoRA fine-tuning to only the first attention layer. Specifically, we disable LoRA in all other layers and retain only the two projection layers, `q_proj` and `k_proj`, in the first attention layer. Furthermore, to make the number of trainable parameters linearly controllable and scalable, we freeze a proportion of the gradients in the initialized `lora_A` and `lora_B` matrices based on a predefined ratio. This allows us to gradually vary the effective amount of the parameter of LoRA.

| Parameter | Value |
|---|---|
| Target layers | `layers.0.self_attn.q_proj,` |
| | `layers.0.self_attn.k_proj` |
| Trainable modules | Attention only |
| LoRA rank ($r$) | 2 |
| LoRA alpha | 16 |
| LoRA dropout | 0.0 |
| Bias setting | None |
| Seed | 3407 |

Table 7: LoRA Configuration

## D   More Experimental Analysis

### D.1   Discussion on the Performance between DARA and LoRA

In this section, we provide a detailed comparison between LoRA and DARA. We focus on both parameter efficiency and performance across various tasks in the TrueMICL.

DARA is specifically designed to enhance a model's ICL ability with minimal parameter overhead. As shown in Figure 5, taking the result of Qwen2-VL on the task **Operator Induction** as an example, when both methods modify the same target—the $k$ and $q$ projection

| Model | Method | VQAv2 | GQA | A-OKVQA | COCO |
|-------|--------|-------|-----|---------|------|
| Qwen2-VL | Zero-shot | 78.6 | 69.9 | 87.8 | 120.1 |
| | No-image | 78.7 | 69.9 | 87.8 | 119.9 |
| | Random | 79.1 | 71.6 | 89.1 | 120.2 |
| | LoRA | 79.3 | 71.3 | 89.9 | 120.1 |
| | DARA | 79.1 | 71.3 | 89.7 | 119.9 |
| Idefics3 | Zero-shot | 62.9 | 65.5 | 79.7 | 117.6 |
| | No-image | 62.7 | 65.6 | 80.0 | 116.9 |
| | Random | 64.1 | 66.4 | 80.8 | 117.6 |
| | LoRA | 64.1 | 66.7 | 80.8 | 116.8 |
| | DARA | 64.2 | 66.5 | 80.7 | 117.3 |
| Phi-3.5-Vision | Zero-shot | 63.2 | 62.9 | 80.3 | 115.7 |
| | No-image | 63.5 | 62.7 | 80.4 | 115.7 |
| | Random | 65.7 | 64.1 | 81.8 | 116.1 |
| | LoRA | 65.3 | 63.9 | 81.5 | 115.7 |
| | DARA | 65.3 | 63.8 | 81.5 | 115.7 |

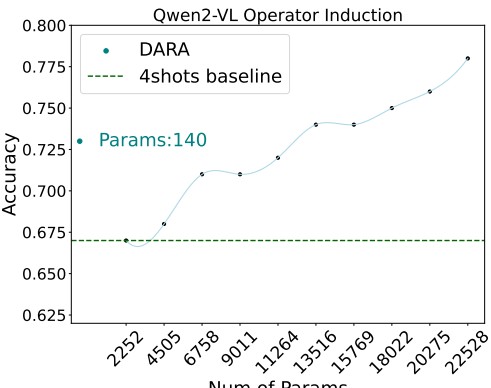

Table 8: Results on four classical vision-language tasks. Accuracy is used for VQAv2, GQA, and A-OKVQA and CIDEr used for COCO. The performance is similar across all methods, suggesting that classical tasks benefit little from in-context learning, and DARA shows no signs of overfitting.

Figure 5: Comparison between DARA (green point) and LoRA (blue curve). DARA outperforms the 4-shot baseline using only 140 parameters, whereas LoRA requires tens of thousands to reach similar performance, highlighting DARA's parameter efficiency.

| Model | Method | Operator | Clock | Outlier | CLEVR | Sudoku | Palindrome |
|-------|--------|----------|-------|---------|-------|--------|------------|
| Qwen2-VL | LoRA | 93.33 | 49.67 | 87.33 | 98.00 | 97.67 | 99.00 |
| | LoRA+DARA | 94.67 (+1.34) | 51.33 (+1.66) | 89.33 (+2.00) | 99.00 (+1.00) | 99.00 (+1.33) | 99.67 (+0.67) |
| Idefics3 | LoRA | 67.67 | 34.00 | 81.33 | 63.00 | 95.00 | 79.67 |
| | LoRA+DARA | 70.00 (+2.33) | 37.00 (+3.00) | 83.00 (+1.67) | 64.33 (+1.33) | 96.00 (+1.00) | 80.33 (+0.66) |
| Phi-3.5-vision | LoRA | 65.33 | 45.67 | 82.00 | 56.33 | 92.67 | 85.33 |
| | LoRA+DARA | 68.00 (+2.67) | 45.00 (-0.67) | 84.00 (+2.00) | 58.00 (+1.67) | 94.33 (+1.66) | 86.33 (+1.00) |

Table 9: Accuracy (%) of different models on six TrueMICL tasks, comparing full-parameter LoRA and LoRA with DARA. While full-parameter LoRA already achieves strong performance across all tasks, the integration of DARA still provides modest but consistent gains.

layers in the first attention layer—DARA significantly outperforms LoRA at low parameter scales. Specifically, LoRA with 2,252 parameters shows no improvement over random 4-shot inference. In contrast, DARA achieves better results using fewer than 225 parameters, matching the performance of LoRA configurations with 11,264 parameters. When applied to the same layer, LoRA needs 40–50 times more parameters than DARA to achieve similar performance. These results demonstrate DARA's effectiveness in low-resource settings while maintaining exceptional parameter efficiency.

We further explore DARA's behavior when combined with full-scale LoRA training. As shown in Table 9, full-parameter LoRA—where millions of parameters are trained—undoubtedly leads to a performance boost across all TrueMICL tasks. However, when DARA is applied on top of full LoRA, we still observe a small but consistent improvement of 1 to 2 percent in accuracy. While this gain may appear small, it is achieved with a parameter cost that is less than one ten-thousandth of that used in full LoRA, once again demonstrating the efficiency and practicality of DARA.

To summarize, DARA shows clear advantages over LoRA in low-parameter regimes, achieving better performance with significantly fewer parameters. It is particularly effective for tasks that truly rely on in-context learning, as shown in TrueMICL. Moreover, DARA remains beneficial even when combined with full-parameter LoRA. These findings demonstrate that DARA serves as both an effective standalone solution and a valuable complement to current adaptation approaches for multimodal in-context learning.

### D.2 Closed-source model performances

We evaluate GPT-4o on the six TrueMICL tasks under both 0-shot and 4-shot settings. As a state-of-the-art multimodal model, GPT-4o's performance can serve as an approximate upper bound. As shown in Table 10, GPT-4o struggles in the 0-shot setting, achieving low accuracy except on Sudoku. Once demonstrations are added, performance improves dramatically across all tasks. This further confirms that TrueMICL requires models to effectively integrate and reason over visual-textual demonstrations, validating its role as a multimodal in-context learning benchmark.

| Model | Method | Operator | Clock | Outlier | CLEVR | Sudoku | Sudoku (hard) | Palindrome |
|-------|--------|----------|-------|---------|-------|--------|----------------|------------|
| **GPT-4o** | Zero-shot | 8.00 | 2.00 | 26.00 | 4.00 | 94.00 | 0.00 | 33.00 |
| | 4-shot MICL | 100.00 | 87.00 | 99.00 | 96.00 | 100.00 | 91.00 | 97.00 |

Table 10: Accuracy (%) of GPT-4o on the six TrueMICL tasks under different settings. Demonstrations greatly improve performance, indicating strong multimodal in-context learning requirements.

### D.3 Amounts of params in DARA

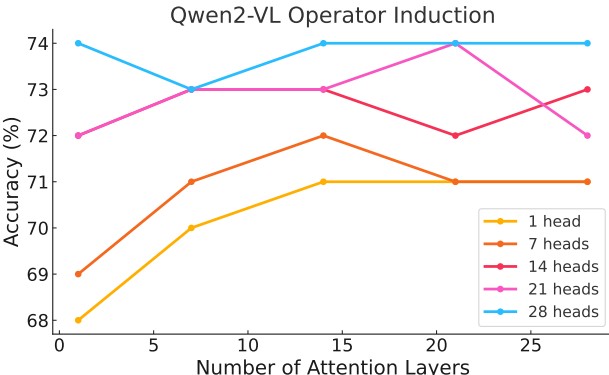

Figure 6: Comparison of performances when applying different settings of changed attention layers and heads.

Figure 6 illustrates the model's performance under different design choices of DARA, where we vary the number of modified layers and attention heads. Taking the $n$-shot setting as an example, the total number of trainable parameters introduced by DARA is calculated as $(n+1) \times$ (number of heads) $\times$ (number of layers). As shown in the figure, within the range of a few hundred parameters, different configurations have no significant difference in the impact on the final performance.

### D.4 Effect of Prompt Design

We provide further analysis on the effect of prompt formulation. Our default prompt is intentionally minimal to avoid leaking visual information into the text and to better test models' ability to reason from image demonstrations alone. To evaluate whether DARA's improvements could instead be attributed to better prompt wording, we compare this with a more explicit instructional prompt.

**Original Prompt (Minimal and Neutral)**

```
System: Learn from the demos and give only the answer.
User:  <demo1> Question:  What is  the  result  of  the  following
```

```
mathematical expression? Answer: 12
(repeated for 4 demos)
<query> Question: What is the result of the following mathematical
expression? Answer:
```

**Instructed Prompt (With Explicit Task Description)**

```
System: The following image-text pairs will show you how to answer
the questions.
From these demos, learn the desired operation on the numbers shown
in each image.
User: Here is the demo image 1: <demo1>
The question is: What is the result of the following mathematical
expression?
The answer is: 12
(repeated for 4 demos)
Here is the query where you need to answer, following the previous
answers:
<query> The question is:  What is the result of the following
mathematical expression?
The answer is:
```

| Model | Prompt | Operator | Clock | Outlier | CLEVR | Sudoku | Palindrome |
|-------|--------|----------|-------|---------|-------|--------|------------|
| Qwen2-VL | Original | 67 | 31 | 87 | 86 | 94 | 96 |
| | Instructed | 67 | 30 | 87 | 87 | 95 | 96 |

Table 11: Performance comparison under two prompt settings (accuracy %). Instructional prompts offer marginal formatting consistency, but little accuracy gain.

**Performance Comparison**   Table 11 shows the performance of Qwen2-VL under both prompt settings. The instructional version yields slightly more consistent formatting, but does not improve overall accuracy. This suggests that the performance gain brought by DARA is not due to prompt wording, but due to its stronger ability to leverage visual support examples.

