# OpenReview forum: "True Multimodal In-Context Learning Needs Attention to the Visual Context"
_colmweb.org/COLM/2025/Conference — COLM 2025_

### Official Review · Reviewer_1vtb · 2025-04-11

**Rating:** 6
**Confidence:** 3
**Ethics Flag:** 1

**Summary:**

This paper investigates the limitation of current Multimodal Large Language Models (MLLMs) in Multimodal In-Context Learning (MICL) where they tend to prioritize textual information over visual cues in demonstration examples ("unimodal trap"). This can hinder genuine multimodal adaptation and task learning for new scenarios.

The authors introduce two contributions aimed at addressing this:
1. Dynamic Attention ReAllocation (DARA): A lightweight fine-tuning strategy (around 160 parameters) designed to encourage the model to attend more to visual tokens within the context images by applying learnable scaling factors to their attention scores.
2. PureMICL: A new benchmark dataset comprising 7 tasks across 4 categories (math, concept binding, pattern finding, novel concept).

These tasks are designed such that solving the query ideally requires understanding the multimodal relationships presented in the demonstrations (task learning), making them purportedly more sensitive to true MICL ability than tasks solvable by text imitation alone. The authors suggest these tasks become difficult if demonstration images are absent.

The paper presents experimental results on several MLLMs (Qwen2-VL, Idefics3, Phi-3.5-Vision, and GPT-4o). The performance on PureMICL for baseline models appears low, suggesting the tasks are challenging. DARA shows some improvement on PureMICL compared to baselines, including parameter-matched LoRA, and demonstrates parameter efficiency. The paper also indicates DARA does not negatively impact performance on standard Vision-Language benchmarks.

**Questions To Authors:**

1. Could you provide a more detailed analysis of the specific instances where DARA leads to the most significant improvements on PureMICL? What characteristics of these tasks might explain this?

2. Have you considered or experimented with applying DARA to transformer layers beyond the first one? What were the preliminary results or what are the challenges in doing so?

3. To further validate the design of PureMICL, could you provide more in-depth ablation studies, perhaps showing the performance drop when specific visual cues in the demonstrations are altered or removed (beyond just removing the entire image)?

4. How sensitive is DARA to the choice of hyperparameters, such as the learning rate or the magnitude of the learnable attention weights? Were these extensively tuned?

**Reasons To Accept:**

1. Addresses a Relevant Problem: The paper tackles the issue of MLLMs potentially neglecting visual information in MICL, which is a recognized challenge in the field.

2. Presents Novel Approaches: The DARA fine-tuning method and the design principles behind the PureMICL benchmark offer new ways to address and evaluate MICL capabilities.

3. Demonstrates Some Empirical Improvements: Experiments show that DARA can lead to performance gains on the proposed PureMICL benchmark compared to certain baselines, suggesting it has some potential to improve visual attention.

4. Highlights Parameter Efficiency: DARA achieves its results with a relatively small number of parameters, which is a desirable characteristic for practical application.

5. Generally Clear Presentation: The paper is reasonably well-written and the concepts are generally understandable.

**Reasons To Reject:**

1. Marginal Performance Gains: While DARA shows improvements, the magnitude of these gains, especially on certain tasks and compared to the potential of full fine-tuning, might be considered relatively modest, placing the contribution marginally above the acceptance threshold.

2. Limited Exploration of DARA: The application of DARA is primarily focused on the first transformer layer. The lack of extensive exploration across different layers might leave questions about its full potential and optimal usage.

3. Convincingness of PureMICL: While the design principles of PureMICL are plausible, further analysis might be needed to definitively demonstrate that the tasks truly and uniquely require deep multimodal understanding of the demonstrations and are not solvable through other means or are significantly impacted by visual perception limitations.

4. Comparison to Other Methods: The comparison to existing methods, while including LoRA, could potentially be broadened to include other recent techniques aimed at improving multimodal understanding in MLLMs.

---

> ### Author Response · Authors · 2025-06-02
> **Response to Reviewer 1vtb 1/3**
>
> Thank you for recognizing the novelty, efficiency, and improved performance of our method, as well as acknowledging the clarity of our presentation. We address each of your questions in detail below.
>
> ### *1. Performance Gains of DARA.*
>
> Thanks for raising this point! ****While DARA’s absolute gains may seem modest in some cases, DARA is efficient, flexible, and brings consistent improvement, making it a practical contribution. We highlight the following key points**:**
>
> 1) **Consistent Gains across Models and Tasks with a few Parameters.** DARA achieves consistent improvements across different models and tasks, using only ~100 trainable parameters and requiring no structural changes. This makes it especially appealing in MICL, where rapid adaptation and parameter efficiency are essential.
>
> 2) **High Flexibility and Extensibility.** DARA is light-weight and can be flexibly integrated with methods like LoRA. It also improves performance when applied on top of full-parameter LoRA (Table 4 in main paper).
>
> 3) **No Trade-of on Standard VL Benchmarks.** DARA maintains strong performance on benchmarks like VQAv2 and COCO (Table 3 in main paper), confirming it does not harm general vision-language capabilities.
>
> In summary, DARA offers a novel and efficient solution to visual ignorance in MICL.
>
> ---
>
> ### *2. More Exploration Ablation on DARA, e.g., adding DARA to more than the 1st layer.*
>
> Thanks for the suggestion! We conducted ablation studies by extending DARA to multiple layers, as shown in [Figure 5]([*https://anonymous.4open.science/r/PureMICL-D1E7/Reviewer4-1vtb-Question2-Different%20Layers%20and%20Heads.png*](https://anonymous.4open.science/r/PureMICL-D1E7/Reviewer4-1vtb-Question2-Different%20Layers%20and%20Heads.png)) in appendix. These experiments demonstrated that applying **DARA to additional layers yields only comparable performance**. One key reason is that **the first layer serves a crucial role in establishing early cross-modal fusion and alignment.** By the second layer, tokens have already accumulated entangled information from other tokens through self-attention, making it harder to differentiate between modalities. Given this trade-off between parameter count and computational cost, we opted to focus on the first layer with several attention heads as our final configuration.
>
> ---
>
> ### *3. Convincingness of PureMICL.*
>
> Thanks for raising this important point. We address it from both empirical and design perspectives as follows:
>
> 1) **Human study validates task design.** We conducted an additional user study with 10 participants to assess whether PureMICL tasks require multimodal context (table below). Without demonstrations (0-shot), participants struggled on most tasks except basic patterns like Sudoku and Palindrome. Performance improved significantly with demonstrations, **confirming that prior knowledge alone is insufficient and multimodal context is essential.**
>
> | **Setting** | **Operator** | **Clock** | **Outlier** | **CLEVR** | **Sudoku** | **Palindrome** |
> | --- | --- | --- | --- | --- | --- | --- |
> | zero-shot | 35 | 32 | 30 | 10 | 97 | 89 |
> | 4-shot | 100 | 100 | 100 | 100 | 100 | 100 |
>
> 2) **PureMICL is unsolvable through other means such as prior knowledge or text hints.** We designed PureMICL to minimize reliance on priors or textual cues.  For example, in the operator induction task, even with arithmetic knowledge and text indicating a math task, the model cannot infer the correct operator without visual cues from the demo images.
>
> 3) **PureMICL focuses on reasoning over multimodal context rather than low-level visual perception.** When designing PureMICL avoids tasks requiring fine-grained visual cues (e.g., counting lines or detecting small changes). Instead, it emphasizes global, semantic patterns such as OCR and object shapes.
>
> 4) **Performance is not limited by visual perception.** Current MLLMs can generally recognize objects, numbers, and layouts when prompted, which suffices for PureMICL. If visual perception were the main bottleneck, increasing demonstrations wouldn’t help—yet performance improves significantly from 0-shot to 4-shot. This suggests the core challenge of PureMICL lies in learning and reasoning over multimodal context, rather than the visual perception.
>
> We thank the reviewer for the comments and will include the discussion in the revision.
>
> ---

---

> > ### Author Response · Authors · 2025-06-02
> > **Response to Reviewer 1vtb 2/3**
> >
> > ### *4. Comparison to Other Methods.*
> >
> > Thanks for raising this point. We would like to address this concern as follows:
> >
> > 1) **Current baselines cover various methods**. We include five baselines, from prompt-based approaches (e.g., random, RICES) to trainable ones like LoRA. While methods like visual instruction tuning are promising, they require heavy fine-tuning. As our goal is to enhance MICL while preserving its lightweight, quick-adaptation nature, we focused on methods aligned with that objective.
> >
> > 2) **We added one more baseline via prompt engineering.** We also experimented with prompt engineering (e.g., using system messages to explicitly instruct the model to attend to demo images). Such strategies yielded similar performance, as models still preferred textual shortcuts. These findings further highlight the need for direct attention rebalancing mechanisms like DARA.
> >
> > | Model | Method | Operator | Clock | Outlier | CLEVR | Sudoku | Palindrome |
> > | --- | --- | --- | --- | --- | --- | --- | --- |
> > | Qwen2-VL | original prompt | 67 | 31 | 87 | 86 | 94 | 96 |
> > |  | instructed prompt | 67 | 30 | 87 | 87 | 95 | 96 |
> >
> > 3) **DARA is flexible and compatible to other methods.** DARA is designed as lightweight and flexible. It is complementary to other approaches such as LoRA and can further boost LoRA’s performance, as shown in Table 4 from the main paper.
> >
> > 4) **DARA’s contribution is conceptually novel.** To our knowledge, DARA is the first method to explicitly address multimodal in-context learning by targeting attention imbalance. This perspective has proven effective through improved performance and is complementary to existing strategies.
> >
> > ### *5. Analysis of the specific instances.*
> >
> > Thanks for the question. To analyze where DARA is most effective, we examined tasks from PureMICL’s Clock Math, which require identifying a consistent arithmetic rule (e.g., +, –, ×) from a few image-based examples and applying it to a new query.
> >
> > A few illustrative examples are shown [here](https://anonymous.4open.science/r/PureMICL-D1E7/Reviewer4-1vtb-Question5-Case%20of%20PureMICL%20advantage.png). In one representative case, “clock_3_11,” the model must infer the transformation between two colored numbers on a clock and apply it to a new pair. Without DARA, models fail by relying on language priors or heuristics, overlooking visual patterns. With DARA, the model focuses on relevant visual cues, infers the rule, and correctly applies it to the query.
> >
> > Another representative case is the outlier detection task. **Without demonstrations, the model cannot determine whether the outlier is defined by color or shape.** Only by observing the demo images can it infer the correct visual criterion. With DARA, the model learns the underlying rule and applies it accurately. These tasks respond well to DARA because, despite their visual simplicity, they require inferring a shared rule—e.g., arithmetic, comparison, or pattern matching—that cannot be derived from text alone.
> >
> > These tasks benefit from DARA because, despite their visual simplicity, **they require inferring shared rules across demo images—like arithmetic, comparison, or pattern matching—that text alone cannot provide.**
> >
> > ### *6. Applying DARA to transformer layers beyond the first block.*
> >
> > Thank you for the question! In short, applying DARA beyond the first block yields similar results, likely due to early-stage cross-modality fusion. We kindly refer you to **the 2nd point above** for details, and we will incorporate this analysis into the main paper in the next revision.
> >
> > ---

---

> > ### Author Response · Authors · 2025-06-02
> > **Response to Reviewer 1vtb 3/3**
> >
> > ### *7. More ablation studies on the design of PureMICL e.g. removing visual cues.*
> >
> > We followed your suggestions and conducted an ablation study by removing visual cues from the demos, as shown in this [figure](https://anonymous.4open.science/r/PureMICL-D1E7/Reviewer4-1vtb-Question7-Without%20vision%20ques.png). In the clock task, arrows and colors originally indicate which two numbers to compute, with prompts like “What is the mathematical result of the image?”
> >
> > We then removed arrows and colors from the images, leaving only digits. In this setting, the model’s accuracy dropped substantially as shown in the table below. **This indicates that features like color and pointer direction are critical**. This controlled ablation highlights the importance of integrating visual and textual information to solve PureMICL tasks.
> >
> > Thank you for this thoughtful suggestion and we will include this new ablation to the next revision!
> >
> > | **zero-shot** | **4-shots** | **4-shots w/o cues** | **DARA 4-shot** | **DARA 4-shot w/o cues** |
> > | --- | --- | --- | --- | --- |
> > | 24% | 31% | 25% | 37% | 25% |
> >
> > ---
> >
> > ### *8. DARA’s hyperparameter sensitivity.*
> >
> > Thanks for the question! **DARA is not highly sensitive to hyperparameter choices**. **We adopted common values without extensive tuning**. The learning rate proved to be the most influential parameter—we used 1e-3, which allowed the learnable attention weights (initialized to 1) to adjust appropriately. While higher learning rates enabled faster but potentially unstable updates, lower rates resulted in insufficient adjustment.
> >
> > The table below summarizes our configuration. These default settings demonstrated consistent performance across various tasks and runs without requiring task-specific tuning. We will include these implementation details in our revision and emphasize that DARA is not hyperparameter sensitive.
> >
> > | **Batch size** | **Epochs** | **Learning rate** | **Warmup steps** | **Optimizer** | **Weight decay** | **Learning rate scheduler** |
> > | --- | --- | --- | --- | --- | --- | --- |
> > | 4 | 5 | 1e-3 | 5 | AdamW | 0.01 | Linear |

---

> > > ### Comment · Reviewer_1vtb · 2025-06-03
> > > **Keeping my score**
> > >
> > > Thank you for your additional experiments --- I will keep my score.

---

> > > > ### Author Response · Authors · 2025-06-03
> > > > **Thank you for keeping the positive score**
> > > >
> > > > Thank reviewer for taking the time to read our response and check the additional experiments. We are glad to hear that reviewer still keeps the positive score.
> > > >
> > > > Thank reviewer again for your valuable time and constructive feedback. We will include our rebuttal details and the suggestions in our paper. Thank you!

---

### Official Review · Reviewer_Ro7V · 2025-04-23

**Rating:** 6
**Confidence:** 4
**Ethics Flag:** 1

**Summary:**

As MLLMs rapidly advance, there is growing interest in their Multimodal In-Context Learning (MICL) capabilities. However, existing MLLMs tend to ignore visual information during MICL and rely heavily on textual cues.
To address this, the paper proposes a fine-tuning method,  called Dynamic Attention ReAllocation (DARA), that introduces additional learnable parameters to rescale the attention weights associated with visual context, encouraging MLLMs to attend more to visual cues.
Moreover, the authors identify a key limitation in existing MICL benchmarks: models can perform well by relying solely on textual information, bypassing the need to interpret visual inputs. To mitigate this, they introduce PureMICL, a new benchmark specifically designed to require integration of both visual and textual modalities. The dataset ensures that successful task completion is only possible through genuine multimodal reasoning.

**Questions To Authors:**

Please refer to the Reasons to Reject section above.

**Reasons To Accept:**

1. The paper addresses an important and timely aspect of MLLMs, specifically focusing on the existing limitations of MICL.
2. The paper is well written and easy to follow.
3. The introduction of the new dataset, PureMICL, is a significant contribution to the community and will likely become a valuable benchmark for evaluating MICL performance in future MLLM research.

**Reasons To Reject:**

1. The paper claims that for any attention scaling factor matrix $F$, there exist corresponding LoRA-style updates $W'_Q$ and $W'_K$ as shown in Eq (5) and Eq (6). However, Eq (6) involves the input matrix $X$, making $W'_K$ dependent on the input. This contradicts the standard LoRA setting, where weight updates are fixed and input-independent. Furthermore, matrix $F$ varies based on the positions of visual tokens in the input sequence, which again violates the assumptions of typical LoRA configurations. Please correct me if I am misunderstanding the formulation.
2. The paper argues that DARA outperforms LoRA in terms of both performance and parameter efficiency. However, the LoRA setup used in the experiments is overly restricted, as it only applies LoRA to the first layer’s  $W_Q$  and  $W_V$  matrices. While this constraint helps match the number of learnable parameters with DARA, it deviates from how LoRA is commonly used in practice. In standard usage, LoRA is applied across all layers and includes $W_V$ with a typical rank of 8, which still offers good parameter efficiency. To make the comparison more fair and informative, the authors should report results for this more representative LoRA configuration. Additionally, comparing DARA against full fine-tuning results should help establish its effectiveness relative to a stronger baseline.
3. Although DARA is intended to boost attention toward visual tokens, the paper does not include any empirical analysis or visualization of the learned attention scaling parameters. Including evidence such as the distribution of these parameters or attention heatmaps would strengthen the claim that DARA effectively encourages the model to focus more on visual information.

---

> ### Author Response · Authors · 2025-06-02
> **Response to Reviewer Ro7v 1/2**
>
> Thank you for recognizing the importance of our motivation and for acknowledging the quality of our writing and the value of our proposed benchmark. We provide detailed responses to each of your concerns below.
>
> ### ***1.** Clarifying the relationship between DARA and LoRA.*
>
> Thanks for this thoughtful observation! We address your concern as follows:
>
> 1) **DARA can be seen as a constrained variant or LoRA rather than a technical equivalent.** The misunderstanding likely arises from viewing DARA as equivalent to LoRA in practice. In fact, as stated in Appendix A, DARA can be theoretically seen as **a special** **constrained variant—not a practical implementation-of LoRA.** The Eq. (6) is a theoretical construct to show that $\mathbf{F}$ can be expressed via low-rank updates. However, as you rightly point out, this requires input-dependent updates and token-position-specific factors, which deviate from standard LoRA assumptions.
>
> 2) **Such constraints provide practical benefit to MICL.** While LoRA is more general, it can be indirect and suboptimal to rebalance the attention allocation for better MICL. In contrast, **DARA's constrained formulation enables targeted visual attention modulation with far fewer parameters**, leading to more efficient MICL improvement.
>
> Thanks again for the pointing this out and we will revise the paper to clarify the theoretical nature of the proof and emphasize DARA’s practical benefits.
>
> ---
>
> ### *2. Using a representative LoRA setting and comparing DARA with full fine-tuning.*
>
> Thanks for raising this point! We address the concern as follows:
>
> 1) **Parameter-matched LoRA reflects low-resource MICL scenarios.** We agree that our LoRA setting deviates from the standard configuration. However, our goal is to simulate **low-resource MICL use cases**, where models should adapt quickly without heavy fine-tuning. Since DARA is very lightweight (~ 100 parameters), we matched LoRA’s parameter scale to ensure a fair efficiency-focused comparison.
>
> 2) **Full LoRA underperforms on unseen tasks.** While more trainable parameters normally improve the task performance, this often reduce generalization to unseen tasks. **We observe that full LoRA exhibits limited transfer ability to untrained tasks, whereas DARA shows relatively better generalization** as shown in Table 9 in Appendix
>
> 3) **DARA complements LoRA.** DARA can be integrated to full-parameter LoRA and **consistently improves MICL performance across tasks and models,** as shown in the table below. This suggests that **DARA's hundreds of parameters significantly influence attention distribution, even when combined with LoRA's millions of parameters.**
>
> | **Model** | **Method** | **Operator** | **Clock** | **Outlier** | **CLEVR** | **Sudoku** | **Palindrome** |
> | --- | --- | --- | --- | --- | --- | --- | --- |
> | Qwen2-VL | LoRA | 93.33 | 49.67 | 87.33 | 98.00 | 97.67 | 99.00 |
> |  | LoRA+DARA | 94.67  (+1.34) | 51.33  (+1.66) | 89.33  (+2.00) | 99.00  (+1.00) | 99.00  (+1.33) | 99.67  (+0.67) |
> | Idefics3 | LoRA | 67.67 | 34.00 | 81.33 | 63.00 | 95.00 | 79.67 |
> |  | LoRA+DARA | 70.00  (+2.33) | 37.00  (+3.00) | 83.00  (+1.67) | 64.33  (+1.33) | 96.00  (+1.00) | 80.33  (+0.66) |
> | Phi-3.5-vision | LoRA | 65.33 | 45.67 | 82.00 | 56.33 | 92.67 | 85.33 |
> |  | LoRA+DARA | 68.00  (+2.67) | 45.00  (-0.67) | 84.00  (+2.00) | 58.00  (+1.67) | 94.33  (+1.66) | 86.33  (+1.00) |
>
> 4) **Full fine-tuning is costly and less practical**. While full fine-tuning may achieve better performance on specific tasks, it brings **significantly computational cost, lower parameter efficiency, and a hier risk of overfitting**, making it less suitable option for fast-adapting MICL use cases.
>
> We appreciate this helpful comments and we will update the paper to include more clarification and discussion.

---

> > ### Author Response · Authors · 2025-06-02
> > **Response to Reviewer Ro7v 2/2**
> >
> > ### ***3.** Empirical Analysis and Visualization of DARA.*
> >
> > Thanks for raising this point! To better understand how DARA affects model behavior, we provide 3 types of visualizations that highlight the effect of DARA in this [figure](https://anonymous.4open.science/r/PureMICL-D1E7/Reviewer3-Ro7V-Question3-Empirical%20Analysis%20and%20Visualization%20of%20DARA.png).
> >
> > 1) **Attention distribution over text vs image tokens (top-left):** Without DARA, the model allocates only 28% of attention to image tokens, focusing primarily on text. With DARA, this rises to 48.7%, indicating a substantial shift toward visual content during response generation.
> >
> > 2) **Attention heatmaps (right):**  The original model shows low attention on demo and query images (dominant blue regions). With DARA, attention over image tokens increases markedly (more red/yellow areas), indicating enhanced sensitivity to visual input. While not all focus aligns with object regions, the overall shift suggests DARA encourages greater incorporation of image information during inference.
> >
> > 3) **Learned  Amplification Factors** **(bottom left)**: We visualize attention scaling factors over four demos and one query image from the first transformer layer (8 heads). Values >1 indicate attention amplification on visual tokens. DARA induces a clear redistribution: (a) demo images consistently receive factors >1, **encouraging stronger reliance on context**; and (b) **different heads specialize**—for example, Head 1 emphasizes Demo 2 (1.27), while Head 5 emphasizes Demo 4 (1.32). These patterns confirm that DARA enables selective, context-aware visual attention during MICL.
> >
> > Thanks again for the suggestion and we will include these analyses in the next revision to better validate the effectiveness of DARA.

---

> > > ### Comment · Reviewer_Ro7V · 2025-06-04
> > >
> > > Thank you for the response. I have raised my score from 5 to 6.

---

> > > > ### Author Response · Authors · 2025-06-04
> > > > **Thank you for raising the score.**
> > > >
> > > > Thank the reviewer for taking the time to read our response. We are delighted that the reviewer has increased the evaluation score! We greatly appreciate the insightful and constructive feedback, and we will incorporate these rebuttal details into our paper. Thank you!

---

### Official Review · Reviewer_gEKm · 2025-05-13

**Rating:** 7
**Confidence:** 4
**Ethics Flag:** 1

**Summary:**

This paper identifies a critical issue in visual language understanding, where Multimodal Large Language Models (MLLMs) over-rely on textual patterns in demonstrations and neglect visual cues (termed the "unimodal trap"). To address this, the authors propose two contributions: 1) Dynamic Attention ReAllocation (DARA), a lightweight fine-tuning method to rebalance attention towards visual tokens , and 2) PureMICL, a new benchmark dataset specifically designed to require understanding of visual context in demonstrations for successful task completion (focusing on "task learning" rather than just "task recognition"). Experiments show current MLLMs struggle with PureMICL, but DARA significantly improves performance on both PureMICL and standard VL tasks.

**Questions To Authors:**

- Does DARA primarily amplify existing relevant visual signals, or could it potentially introduce biases by forcing attention onto less relevant visual tokens in some cases?
- Beyond the tasks in PureMICL, what other complex, real-world scenarios do you foresee benefiting most significantly from overcoming the "unimodal trap" using methods like DARA?

**Reasons To Accept:**

- This paper addresses a significant and widely recognized limitation (visual neglect) in current MLLMs' capabilities.
- This study proposes DARA, a parameter-efficient fine-tuning technique that demonstrably improves MICL performance by encouraging visual attention. It outperforms comparable LoRA configurations significantly in low-parameter regimes.
- PureMICL is contributed as a diagnostic dataset specifically designed to test and necessitate true multimodal understanding in context, moving beyond limitations of standard VL benchmarks for MICL evaluation.
- Comprehensive experiments are provided, across multiple MLLMs (Qwen2-VL, Idefics3, Phi-3.5-Vision), showing the shortcomings of existing models on PureMICL and the consistent effectiveness of DARA.

**Reasons To Reject:**

- While DARA aims to induce stronger attention to visual tokens via learnable reweighting, the paper lacks sufficient evidence that this actually occurs. To validate DARA's effectiveness, the authors should provide visualizations or quantitative analyses of attention maps/statistics, demonstrating the shift in attention towards visual tokens after tuning.
- The paper lacks a baseline that would provide a clearer understanding of DARA's advantage: a hard-coded setting where specific attention heads are forced to attend only to visual tokens. This simple comparison would isolate the benefit of DARA's learned, dynamic reweighting versus a static constraint.
- The paper does not provide performance data for state-of-the-art VLMs (ChatGPT Vision, Claude, Gemini). This raises the question of whether scaling model size or pre-training data alone could solve some of the PureMICL tasks, potentially negating the need for a novel method like DARA.
- PureMICL Task Diversity: While PureMICL introduces tasks requiring visual context, the task types (math, concept binding, patterns, novel concepts) might not fully represent the breadth of real-world multimodal applications.
- Comparison with Other Mitigation Strategies: The paper compares DARA primarily to LoRA and baseline MICL. Discussion or comparison with other potential strategies for mitigating visual neglect (e.g., different model architectures, prompt engineering techniques) is limited.

---

> ### Author Response · Authors · 2025-06-02
> **Response to Reviewer gEKm 1/3**
>
> Thank you for recognizing our motivation, DARA’s performance, the value of PureMICL, and the scope of our experiments! Below, we address each of your concerns in detail.
>
> ### ***1.** Visualization and Empirical Analysis of DARA.*
>
> Thank you for the suggestion. We have added two additional analyses as shown in this [figure](https://anonymous.4open.science/r/PureMICL-D1E7/Reviewer2-gEKm-Question1-Visualization%20and%20Empirical%20Analysis%20of%20DARA.png):
>
> 1) **Attention distribution of text vs. image tokens (left).** Without DARA, models focus primarily on text tokens, giving only 28% of the attention to image content. After applying DARA, we observe that attention to image tokens significantly increases to 48.7%,  indicating a substantial shift toward visual content during response generation.
>
> 2) **Attention heatmaps over demo and query images (right).**  Attention heatmap of input images further show that, without DARA, both demonstration and query images do not receive much attention, as indicated by predominantly blue regions. After tuning with DARA, **attention intensifies over relevant image areas** (more orange/red), confirming improved visual focus.
>
> Thanks again and we will include these analyses in the next revision.
>
> ---
>
> ### ***2.** Hard-coded Attention Adjustment as another baseline.*
>
> Thank you for the suggestion! We address this point as follows.
>
> 1) **Hard-coded attention adjustment disrupts learned priors.** As each attention head is with certain learned priors from the pre-training, **hard-coded modification can lead to degraded performance.** We followed your suggestion and implemented a hard-coded variant where we forced specific attention heads to focus exclusively on visual tokens. Concretely, we selected the first half of the attention heads in a layer and masked out text-token positions in two settings: (1) only at the first layer, and (2) across the first 8 layers. **In both cases, performance dropped to near-random, suggesting that rigid constraints disrupt the pre-trained roles of individual heads.**
>
> 2) **DARA enables adaptive attention modulation.** Unlike manual masking, DARA introduces **learnable visual amplification factors.** This allows the model to adaptively reweight attention to image tokens without discarding text, thus lead to better MICL performance.
>
> 3) **Future work: training-free attention adjustments.** We agree that how to better understand the learned priors in each head and how to design a training-free attention adjustment is a very interesting direction for future work.
>
> Thanks again for your thoughtful suggestion and we will include these discussions in the next revision.
>
> ---
>
> ### ***3.** Performance on SOTA VLMS and the necessity of DARA.*
>
> | **Model** | **Method** | **Operator** | **Clock** | **Outlier** | **CLEVR** | **Sudoku** | **Sudoku (hard)** | **Palindrome** |
> | --- | --- | --- | --- | --- | --- | --- | --- | --- |
> | GPT-4o | zero-shot | 8.00 | 2.00 | 26.00 | 4.00 | 94.00 | 0.00 | 33.00 |
> |  | MICL | 100.00 | 87.00 | 99.00 | 96.00 | 100.00 | 91.00 | 97.00 |
>
> Thank you for this important question. We evaluated GPT-4o and observed stronger performance (see table above). However, this does not diminish the value of DARA, for the following reasons.
>
> 1) **It is difficult to attribute improvements solely to model size or data scale.** Closed models lack transparency, making it difficult to attribute their gains solely to scale. Their undisclosed architecture, data, and fusion strategies prevent us from isolating the source of performance improvements.
>
> 2) **Scaling alone still does not solve the problem completely**. When we increased the difficulty of PureMICL tasks (e.g., we designed a hard version of Sudoku as shown [here](https://anonymous.4open.science/r/PureMICL-D1E7/Reviewer2-gEKm-Question3-Caseshow%20of%20SOTA%20model.png), GPT-4o’s accuracy dropped significantly, suggesting that **the core challenge of** **MICL persists even at scale**.
>
> 3) **DARA is not in conflict with scaling**. Its lightweight design allows integration during or after training. Our results (Table 4 in the main paper) show that **DARA further improves performance when combined with LoRA,** and could similarly benefit models like GPT-4o.
>
> 4) **DARA offers a practical and effective way to enhance MICL ability for resource-constrained open models.** Though these models cannot match GPT-4o’s scale, they are also widely adopted in practice and the MICL improvement from DARA is still critical for their downstream applications.
>
> 5) **DARA’s contribution is conceptually novel**. To our knowledge, DARA is the first to improve MICL by targeting attention imbalance. This perspective has proven effective through improved performance and is complementary to existing strategies.
>
> We believe these reasons make DARA still a valuable and complementary contribution to the community. Thanks again and we will add the results and discussions to the main paper in the next revision.
>
> ---

---

> > ### Author Response · Authors · 2025-06-02
> > **Response to Reviewer gEKm 2/3**
> >
> > ### ***4.** Task Diversity of PureMICL.*
> >
> > Thanks for the thoughtful comment! We would like to address your concerns in 3 parts.
> >
> > 1) **PureMICL provides** **coverage over core MICL ability.** The current task types (e.g., math reasoning, concept binding, novel concepts) are chosen to **directly reflect core capabilities required in real-world in-context applications.** PureMICL already reveals significant limitations in current MLLMs and the value of DARA.
> >
> > 2) **PureMICL is a scalable and extensible framework.** PureMICL is also **a configurable generation framework**, allowing easy expansion to new domains and difficulty levels. This makes it accessible for the community to extend and adapt the benchmark to different applications or knowledge domains.
> >
> > 3) **PureMICL as a diagnostic criteria for MICL.** We agree that broader real-world coverage is important. However, defining a “complete” set of multimodal knowledge remains an open challenge. PureMICL focuses on precise diagnosis, serving as a first step toward more faithful MICL evaluation. It successfully reveals key blind spots in current MLLMs and the value of methods like DARA.
> >
> > ---
> >
> > ### ***5.** Comparison with strategies like prompt engineering and model architectures.*
> >
> > Thanks for raising this point. We address this concern as follows.
> >
> > 1) **Prompt engineering yields slightly different output but limited improvement.** Following your suggestion, we tested prompts with explicit instructions, such as “pay attention to the demo images” and emphasis on output formatting. While such prompts slightly affect textual aspects like formatting, **overall performance remains nearly unchanged across PureMICL tasks (table below).** This confirms that prompt-based methods alone are insufficient to address the unimodal trap. We will include the prompt details and discussion in the revised version.
> >
> > | **Model** | **Method** | **Operator** | **Clock** | **Outlier** | **CLEVR** | **Sudoku** | **Palindrome** |
> > | --- | --- | --- | --- | --- | --- | --- | --- |
> > | Qwen2-VL | original prompt | 67 | 31 | 87 | 86 | 94 | 96 |
> > |  | prompt with detailed instructions | 67 | 30 | 87 | 87 | 95 | 96 |
> >
> > 2) **Current experiments cover diverse architectures.** We evaluate DARA on three open-source MLLMs—Qwen2-VL, Idefics3, and Phi-3.5-Vision—**spanning** **different visual encoders (e.g., CLIP, SigLIP), and language backbones**. This demonstrates that DARA generalizes well across existing MLLM types without architectural assumptions.
> >
> > 3) **DARA is model-agnostic and easy to integrate.** Since DARA is lightweight and operates directly on attention scores, it can be applied to any transformer-style model **without changing the architecture or pre-training setup**, **making it broadly usable.**
> >
> > 4) **Beyond current structure, DARA is compatible with emerging attention-based architectures.** Our experiments focus on image-to-text MLLMs with a standard visual-projection-LLM setup (e.g., Qwen2-VL). However, DARA’s **lightweight design makes it readily applicable to any attention-based model**, including unified MLLMs with interleaved multimodal inputs and outputs. Extending DARA to newer architectures is a promising direction for future work.

---

> > > ### Author Response · Authors · 2025-06-02
> > > **Response to Reviewer gEKm 3/3**
> > >
> > > ### *6. Does DARA enhance useful visual cues or risk biasing toward irrelevant ones?*
> > >
> > > Thanks for raising this point! We address your concerns in 3 parts:
> > >
> > > 1) **DARA's primary purpose is to rebalance attention between text and image tokens.** Rather than seeking better visual features within images, it helps reduce the model's overreliance on text by encouraging greater attention to visual information.
> > >
> > > 2) **DARA shows good cross-task transfer and retain performance on standard VL tasks.**  DARA adjusts attention by learning from the task, naturally aligning it with task-relevant visual cues. However, DARA does not **overfit to spurious patterns** as our cross-task transfer results (Table 9 in the appendix) show that DARA-trained models retain performance even outside the training task. Moreover, **DARA does not degrade the performance on standard VL tasks** such as VQAv2 and image caption as shown in Table 3 in the main paper.
> > >
> > > 3) **Task-specific adaptation is expected but does not diminish DARA’s value.** As with any fine-tuning method, some task-specific adaptation is expected. However, this is outweighed by DARA’s key strength, i.e., **reducing the over-reliance on text tokens and enabling better leverage visual context**. This rebalancing is crucial for realizing MICL’s full potential, as shown by consistent gains on PureMICL tasks.
> > >
> > > Thanks again for your valuable question and we will add the discussion in the next revision!
> > >
> > > ---
> > >
> > > ### *7. Real-world MICL settings that can benefit from DARA.*
> > >
> > > Thank you for this valuable question! Using DARA to overcome the unimodal trap in MICL holds significant potential in complex real-world applications. Examples include:
> > >
> > > 1) **GUI Agents**. MLLM-based GUI agents need to adapt to new software and workflows by interpreting user-provided multimodal demos—e.g., screenshots with instructions. If they overlook the demo screenshots, they cannot truly acquire new skills—severely limiting their practical applications.
> > >
> > > 2) **Automatic Scientific Discovery.** Tasks like protein structure prediction require integrating chemical diagrams with written instructions. Without effective learning from multimodal demonstrations, models may miss crucial patterns needed for accurate prediction and discovery.
> > >
> > > 3) **Medical Foundation Models.**  Patient-specific adaptation in medical domains relies heavily on multimodal demos, e.g., medical images with clinical notes from previous related patients. To deliver accurate medical guidance, models must develop genuine MICL capabilities, so that to adapt effectively from multimodal examples.

---

> > > > ### Author Response · Authors · 2025-06-06
> > > >
> > > > Thank you for your valuable feedback. As the discussion period is underway, we just wanted to kindly check whether our additional experiments and responses have addressed your concerns. We’re happy to clarify any remaining questions you may have. Thanks again!

---

> > > > > ### Comment · Reviewer_gEKm · 2025-06-08
> > > > >
> > > > > I appreciate the detailed explanation, and the rating has been updated to reflect this.

---

> > > > > > ### Author Response · Authors · 2025-06-08
> > > > > > **Thank you for raising the score**
> > > > > >
> > > > > > Thank you for taking the time to read our response. We are truly glad to see that it addressed your concerns, and we greatly appreciate the increased score. We are also grateful for your thoughtful and constructive feedback and will incorporate the discussion into the final version. Thanks again!

---

### Official Review · Reviewer_Lzs5 · 2025-05-16

**Rating:** 7
**Confidence:** 3
**Ethics Flag:** 1

**Summary:**

The paper argues that current MLLMs do not make full use of the visual information in demonstrations and that prior MICL test sets fail to stress this aspect. To address this gap, it introduces a new MICL dataset, PureMICL, which evaluates an MLLM’s ability to integrate multimodal information. In addition, it proposes Dynamic Attention Reallocation (DARA), a simple and efficient fine-tuning strategy that pushes the model to pay more attention to image cues.

**Questions To Authors:**

Another paper[1] is similar in topic but focuses more on text-image pairs, which may be a useful reference.

[1] Zeng, Yuchen, Wonjun Kang, Yicong Chen, Hyung Il Koo, and Kangwook Lee. "Can MLLMs Perform Text-to-Image In-Context Learning?." In First Conference on Language Modeling.

**Reasons To Accept:**

The paper’s discussion of the weakness in current MLLM and MICL benchmarks is convincing and meaningful. The PureMICL benchmark provides several fresh directions for assessing MICL, which can guide future MLLM development toward stronger multimodal skills. The proposed fine-tuning method is simple and effective, and its comparison with several baselines soundly supports the authors’ claims.

**Reasons To Reject:**

The paper may omit some experimental details. Here are the main issues I observed:

1. Prompt design is under-specified. All models used in the paper are dialogue-style, yet the paper does not show the prompts used when running PureMICL tasks. I assume the experiments feed the models plain image-text pairs and then a stand-alone query image. If so, the chosen MLLMs may not be familiar with such an MICL format and could struggle to output answers that exactly match the dataset’s required wording. Under this setup, DARA may have an extra advantage because it is pre-exposed to support data.

    For a fair comparison, here are some possible prompt design for reference:
    - add an initial instruction stating that several image-text pairs will follow
    - add a final reminder asking the model to produce an answer in the same style as the context
    - optionally state the concrete task at the very start, which would form another strong baseline

2. Definition of the "No-image" baseline is unclear.  Does it supply only the answer text for every demo, or does it also include textual descriptions of the demo images?

3. In the main results table (Table 2), you may wang to add a final “Overall” column that reports the average performance across the seven tasks, so that the gaps between baselines are easier to see.

4. Are LoRA and DARA evaluated on top of the Random baseline? You may want to state more clearly in the paper.

---

> ### Author Response · Authors · 2025-06-02
> **Response to Reviewer Lzs5 1/2**
>
> We thank the reviewer for recognizing the proposed benchmark as “convincing and meaningful” and the proposed method as “simple and effective”. We provide detailed responses for each concern below.
>
> ### ***1.** Specify the prompt design and add more information to the prompt for comparison.*
>
> Thank you for raising this point! Our text prompts are designed to be simple and minimal to avoid leakage of visual content and to examine the visual understanding ability. For instance, the *operator induction* task adopts the following **original prompt**:
>
> ```bash
> System: Learn from the demos and give the answer based on the query.
> User: <demo1> Question: What is the result of the following mathematical expression? Answer: 12
> … (repeated for 4 demos)
> <query> Question: What is the result of the following mathematical expression? Answer:
> ```
> where <demo1> and <query> specify the positions of the corresponding images.
> To see whether models benefit from more explicit prompting, we followed your suggestions and conducted additional experiments using **stronger instruction cues**, stating the concrete tasks, announcing the upcoming examples, and reminding the model to follow the answer style. This revised **prompt with detailed instructions is:**
>
> ```bash
> System: The following image-text pairs demonstrate a consistent mathematical operation applied to numbers displayed in the images. Your task is to carefully observe each example and identify the underlying rule. Then, apply this rule to the query image to compute the correct answer. Pay close attention to how the input numbers are presented and how the answers are derived in each case.
> User: Here is the demo image 1: <demo1>
> The question is: What is the result of the following mathematical expression?
> The answer is: 12
> (repeated for 4 demos)
> Here is the query where you need to answer, following the previous answers:
> <query> The question is: What is the result of the following mathematical expression?
> The answer is:
> ```
>
> Our experiments show that **different prompts can produce slightly different outputs and performance**, as shown in the table below. However, **these prompt variations do not significantly improve overall MICL performance**. While carefully crafted text prompts can help guide response formatting, **they offer limited improvement in visual understanding capabilities and therefore cannot significantly enhance performance on PureMICL**. Besides, as DARA is orthogonal to manual prompt design, they can be applied together in applications. Thanks for this feedback and we will add all the detailed prompts for each task in the next revision.
>
> | **Model** | **Method** | **Operator** | **Clock** | **Outlier** | **CLEVR** | **Sudoku** | **Palindrome** |
> | --- | --- | --- | --- | --- | --- | --- | --- |
> | Qwen2-VL | original prompt | 67 | 31 | 87 | 86 | 94 | 96 |
> |  | prompt with detailed instructions | 67 | 30 | 87 | 87 | 95 | 96 |
>
> ---
>
> ### ***2.** Definition of the “No-image” baseline.*
>
> Thanks for your question. The “No-image” baseline is designed to highlight the indispensable role of visual information in MICL, ensuring that the query cannot be correctly answered without referring to the demo images. Specifically, **we remove all demonstration images without replacing them with textual descriptions.** Each demonstration thus contains only the textual question and answer, followed by the query image and its question.
>
> Replacing images with textual descriptions is certainly a possible alternative and the results largely depend on the level of detail in the descriptions. When the descriptions are general or vague (e.g., “an image with numbers”), they provide little to no benefit. In contrast, highly specific descriptions can lead to improved performance, such as those explicitly listing OCR results (e.g., “an image with a number of 2, 3, and a question mark in the middle”). Because we aim to show the task is unsolvable without looking at demo images, the “No-image” baseline does not include the textual descriptions.
>
> We appreciate the reviewer’s insightful comment and will clarify this point more explicitly in the revised version.
>
> ---
>
> ### ***3.** Add an “Overall” Column.*
>
> | **Model** | **Zero-Shot** | **No-image** | **Random** | **RICES** | **LoRA** | **DARA** |
> | --- | --- | --- | --- | --- | --- | --- |
> | **Qwen2-VL** | 36.38 | 41.00 | 77.67 | 79.71 | 80.24 | **83.00** |
> | **Idefics3** | 25.95 | 29.05 | 41.95 | 43.43 | 45.38 | **48.67** |
> | **Phi-3.5-vision** | 23.00 | 27.86 | 44.62 | 47.05 | 48.67 | **52.81** |
>
> Thank you very much for this valuable suggestion! We put the average performance across tasks above. The overall performance also highlights the performance boost from DARA compared with baselines. We will update the main experimental table in the revision.
>
> ---

---

> > ### Author Response · Authors · 2025-06-02
> > **Response to Reviewer Lzs5 2/2**
> >
> > ### ***4.** Are LoRA and DARA on Random Baseline?*
> >
> > Thank you for your question. In our experiments, **both LoRA and DARA are built on the random baseline, as RICES’s performance is comparable to the random baseline** (Table 2 in the main paper). This is primarily because each task in our dataset is generated according to consistent rules, and the corresponding images across tasks are highly similar in both content and structure. **Therefore, selecting demonstrations based on similarity does not bring much benefit,** and we use the random baseline for LoRA and DARA, which also avoids the additional computational cost for RICES.
> >
> > Thanks again for pointing this out and we will make this design choice more explicit in the revised version.
> >
> > ---
> >
> > ### ***5.** Discuss a related paper.*
> >
> > Thank you for the suggestion! Zeng [](http://et.al/)et al. (2024) investigated text-to-image in-context learning capabilities of MLLMs and introduced CoBSAT, a comprehensive benchmark. Their work revealed significant challenges that MLLMs face with T2I-ICL tasks. In comparison, our work examines multimodal ICL. Through PureMICL, we demonstrate **similar limitations in current models' ICL capabilities as CoBSAT** and present DARA as a solution to enhance MICL performance. We will include this relevant work in our revision.

---

> > > ### Author Response · Authors · 2025-06-06
> > >
> > > Thank you for your valuable feedback. As the discussion period is underway, we just wanted to kindly check whether our additional experiments and responses have addressed your concerns. We’re happy to clarify any remaining questions you may have. Thanks again!

---

> > > > ### Comment · Reviewer_Lzs5 · 2025-06-08
> > > >
> > > > Thank you for the thorough response, especially for conducting many new experiments to address the concerns raised by me and other reviewers. I have carefully reviewed the rebuttal and am willing to raise my score from 6 to 7.

---

> > > > > ### Author Response · Authors · 2025-06-09
> > > > > **Thank you for your thoughtful follow-up and score update**
> > > > >
> > > > > Thank you for taking the time to carefully review our response. We truly appreciate your positive update and are glad to hear that the additional experiments and clarifications addressed your concerns. We will incorporate the relevant discussion into the final version. Thank you again!

---

### Author Response · Authors · 2025-06-10
**Discussion Phase Summary**

We sincerely thank all reviewers for their thoughtful feedback and active engagement throughout the review and discussion phases. We are especially grateful for the recognition of the importance of addressing visual neglect in MICL *(gEKm, Ro7V, 1vtb)*, the efficiency and practicality of DARA *(Lzs5, gEKm, 1vtb)*, the depth of our experimental results *(Lzs5, gEKm)*, the value of the PureMICL benchmark *(Lzs5, gEKm, Ro7V)*, and the clarity of our presentation *(Ro7V, 1vtb)*. **We are glad our additional experiments and clarifications during the discussion period addressed the remaining concerns, and appreciate that all reviewers recommend acceptance.**

As the discussion phase concludes, we would like to summarize the key additional analyses that address common themes raised in the reviews, including:

- **Empirical analyses confirm DARA's impact on visual attention.** Attention distribution and spatial attention heatmaps show DARA consistently increases focus on image tokens *(gEKm, Ro7V)*.
- **DARA generalizes across tasks and complements other methods.** DARA can also improve performance on untrained tasks, integrates well with LoRA for better performance, and requires only ~100 parameters *(gEKm, Ro7V).*
- **Prompt engineering and SOTA models are insufficient for PureMICL.** Instructed prompts alone cannot significantly improve the performance on PureMICL; GPT-4o fails on harder tasks from PureMICL, confirming the need for MICL-specific solutions like DARA *(Lzs5, gEKm, 1vtb).*
- **Human studies and ablations support the design of PureMICL.** Human studies and visual cue removals validate that solving PureMICL tasks requires truly attending to the multimodal context *(1vtb)*.
- **Additional Analyses and Clarifications.** We have also provided additional clarifications addressing individual concerns, including: clarification of the prompt and baseline settings *(Lzs5)*; explanation of DARA’s relationship to LoRA *(Ro7V)*; ablation study on DARA’s placement across layers *(1vtb)*; evaluation of hard-coded attention as a control baseline *(gEKm).*

We will incorporate these experimental results and clarifications during rebuttal into the final version. Thank you again for your time and consideration.

---

### Decision · Program_Chairs · 2025-07-08

**Decision:**

Accept

**Comment:**

This paper introduces a new method and a benchmark to address the "unimodal trap" where multimodal models ignore visual information in context. The reviewers all agreed that this is an important problem. They praised the new PureMICL benchmark as a valuable contribution. Initially, the reviewers expressed some reservations. They questioned if the proposed DARA method actually shifted the model's attention and asked for more experiments to validate the benchmark's design. The preliminary ratings were generally positive but reflected these concerns.

The authors provided a comprehensive response with many new experiments. They included visualizations that showed DARA successfully redirects attention to visual information. They also added human studies and further ablation experiments to strengthen the claims about their benchmark. These additions satisfied the reviewers' concerns. All reviewers found the rebuttal convincing and updated their reviews to recommend acceptance. From the AC's perspective, while the methodology is interesting, the current examples may not be comprehensive or robust enough, especially against large-scale models. However, the novel methodology and important problem setting are interesting enough to warrant acceptance.